# Matrix Completion with Hierarchical Graph Side Information

**Adel Elmahdy**[*]
ECE, University of Minnesota
adel@umn.edu

**Junhyung Ahn**[*]
EE, KAIST
tonyahn96@kaist.ac.kr

**Changho Suh**
EE, KAIST
chsuh@kaist.ac.kr

**Soheil Mohajer**
ECE, University of Minnesota
soheil@umn.edu

## Abstract

We consider a matrix completion problem that exploits social or item similarity graphs as side information. We develop a universal, parameter-free, and computationally efficient algorithm that starts with hierarchical graph clustering and then iteratively refines estimates both on graph clustering and matrix ratings. Under a hierarchical stochastic block model that well respects practically-relevant social graphs and a low-rank rating matrix model (to be detailed), we demonstrate that our algorithm achieves the information-theoretic limit on the number of observed matrix entries (i.e., optimal sample complexity) that is derived by maximum likelihood estimation together with a lower-bound impossibility result. One consequence of this result is that exploiting the *hierarchical* structure of social graphs yields a substantial gain in sample complexity relative to the one that simply identifies different groups without resorting to the relational structure across them. We conduct extensive experiments both on synthetic and real-world datasets to corroborate our theoretical results as well as to demonstrate significant performance improvements over other matrix completion algorithms that leverage graph side information.

## 1 Introduction

Recommender systems have been powerful in a widening array of applications for providing users with relevant items of their potential interest [1]. A prominent well-known technique for operating the systems is low-rank matrix completion [2–18]: Given partially observed entries of an interested matrix, the goal is to predict the values of missing entries. One challenge that arises in the big data era is the so-called *cold start problem* in which high-quality recommendations are not feasible for new users/items that bear little or no information. One natural and popular way to address the challenge is to exploit other available side information. Motivated by the social homophily theory [19] that users within the same community are more likely to share similar preferences, social networks such as Facebook's friendship graph have often been employed to improve the quality of recommendation.

While there has been a proliferation of social-graph-assisted recommendation algorithms [1, 20–40], few works were dedicated to developing theoretical insights on the usefulness of side information, and therefore the maximum gain due to side information has been unknown. A few recent efforts have been made from an information-theoretic perspective [41–44]. Ahn et al. [41] have identified the maximum gain by characterizing the optimal sample complexity of matrix completion in the presence of graph side information under a simple setting in which there are two clusters and users

---

[*]Equal contribution. Corresponding author: Changho Suh.

within each cluster share the same ratings over items. A follow-up work [42] extended to an arbitrary number of clusters while maintaining the same-rating-vector assumption per user in each cluster. While [41, 42] lay out the theoretical foundation for the problem, the assumption of the single rating vector per cluster limits the practicality of the considered model.

In an effort to make a further progress on theoretical insights, and motivated by [45], we consider a more generalized setting in which each cluster exhibits another sub-clustering structure, each sub-cluster (or that we call a "group") being represented by a different rating vector yet intimately-related to other rating vectors within the same cluster. More specifically, we focus on a hierarchical graph setting wherein users are categorized into two clusters, each of which comprises three groups in which rating vectors are broadly similar yet distinct subject to a linear subspace of two basis vectors.

**Contributions:** Our contributions are two folded. First we characterize the information-theoretic sharp threshold on the minimum number of observed matrix entries required for reliable matrix completion, as a function of the quantified quality (to be detailed) of the considered hierarchical graph side information. The second yet more practically-appealing contribution is to develop a computationally efficient algorithm that achieves the optimal sample complexity for a wide range of scenarios. One implication of this result is that our algorithm fully utilizing the *hierarchical graph* structure yields a significant gain in sample complexity, compared to a simple variant of [41, 42] that does not exploit the relational structure across rating vectors of groups. Technical novelty and algorithmic distinctions also come in the process of exploiting the hierarchical structure; see Remarks 2 and 3. Our experiments conducted on both synthetic and real-world datasets corroborate our theoretical results as well as demonstrate the efficacy of our proposed algorithm.

**Related works:** In addition to the initial works [41, 42], more generalized settings have been taken into consideration with distinct directions. Zhang et al. [43] explore a setting in which both social and item similarity graphs are given as side information, thus demonstrating a synergistic effect due to the availability of two graphs. Jo et al. [44] go beyond binary matrix completion to investigate a setting in which a matrix entry, say $(i, j)$-entry, denotes the probability of user $i$ picking up item $j$ as the most preferable, yet chosen from a known finite set of probabilities.

Recently a so-called *dual* problem has been explored in which clustering is performed with a partially observed matrix as side information [46, 47]. Ashtiani et al. [46] demonstrate that the use of side information given in the form of pairwise queries plays a crucial role in making an NP-hard clustering problem tractable via an efficient k-means algorithm. Mazumdar et al. [47] characterize the optimal sample complexity of clustering in the presence of similarity matrix side information together with the development of an efficient algorithm. One distinction of our work compared to [47] is that we are interested in both clustering and matrix completion, while [47] only focused on finding the clusters, from which the rating matrix cannot be necessarily inferred.

Our problem can be viewed as the prominent low-rank matrix completion problem [1–4, 6–18] which has been considered notoriously difficult. Even for the simple scenarios such as rank-1 or rank-2 matrix settings, the optimal sample complexity has been open for decades, although some upper and lower bounds are derived. The matrix of our consideration in this work is of rank 4. Hence, in this regard, we could make a progress on this long-standing open problem by exploiting the structural property posed by our considered application.

The statistical model that we consider for theoretical guarantees of our proposed algorithm relies on the Stochastic Block Model (SBM) [48] and its hierarchical counterpart [49–52] which have been shown to well respect many practically-relevant scenarios [53–56]. Also our algorithm builds in part upon prominent clustering [57, 58] and hierarchical clustering [51, 52] algorithms, although it exhibits a notable distinction in other matrix-completion-related procedures together with their corresponding technical analyses.

**Notations:** Row vectors and matrices are denoted by lowercase and uppercase letters, respectively. Random matrices are denoted by boldface uppercase letters, while their realizations are denoted by uppercase letters. Sets are denoted by calligraphic letters. Let $\mathbf{0}_{m \times n}$ and $\mathbf{1}_{m \times n}$ be all-zero and all-one matrices of dimension $m \times n$, respectively. For an integer $n \geq 1$, $[n]$ indicates the set of integers $\{1, 2, \ldots, n\}$. Let $\{0, 1\}^n$ be the set of all binary numbers with $n$ digits. The hamming distance between two binary vectors $u$ and $v$ is denoted by $d_{\mathrm{H}}(u, v) := \|u \oplus v\|_0$, where $\oplus$ stands for modulo-2 addition operator. Let $\mathbb{1}[\cdot]$ denote the indicator function. For a graph $\mathcal{G} = (V, E)$ and two disjoint subsets $X$ and $Y$ of $V$, $e(X, Y)$ indicates the number of edges between $X$ and $Y$.

## 2 Problem Formulation

**Setting:** Consider a rating matrix with $n$ users and $m$ items. Each user rates $m$ items by a binary vector, where $0/1$ components denote "dislike"/"like" respectively. We assume that there are two clusters of users, say $A$ and $B$. To capture the low-rank of the rating matrix, we assume that each user's rating vector within a cluster lies in a linear subspace of *two* basis vectors. Specifically, let $v_1^A \in \mathbb{F}_2^{1 \times m}$ and $v_2^A \in \mathbb{F}_2^{1 \times m}$ be the two linearly-independent basis vectors of cluster $A$. Then users in Cluster $A$ can be split into three groups (e.g., say $G_1^A$, $G_2^A$ and $G_3^A$) based on their rating vectors. More precisely, we denote by $G_i^A$ the set of users whose rating vector is $v_i^A$ for $i = 1, 2$. Finally, the remaining users of cluster $A$ from group $G_3^A$, and their rating vector is $v_3^A = v_1^A \oplus v_2^B$ (a linear combination of the basis vectors). Similarly we have $v_1^B, v_2^B$ and $v_3^B = v_1^B \oplus v_2^B$ for cluster $B$. For presentational simplicity, we assume equal-sized groups (each being of size $n/6$), although our algorithm (to be presented in Section 4) allows for any group size, and our theoretical guarantees (to be presented in Theorem 2) hold as long as the group sizes are order-wise same. Let $M \in \mathbb{F}^{n \times m}$ be a rating matrix wherein the $i^{\text{th}}$ row corresponds to user $i$'s rating vector.

We find the Hamming distance instrumental in expressing our main results (to be stated in Section 3) as well as proving the main theorems. Let $\delta_g$ be the normalized Hamming distance among distinct pairs of *group's* rating vectors within the same cluster: $\delta_g = \frac{1}{m} \min_{c \in \{A, B\}} \min_{i,j \in [3]} d_{\text{H}}\left(v_i^c, v_j^c\right)$. Also let $\delta_c$ be the counterpart w.r.t. distinct pairs of rating vectors *across different clusters*: $\delta_c = \frac{1}{m} \min_{i,j \in [3]} d_{\text{H}}\left(v_i^A, v_j^B\right)$, and define $\delta := \{\delta_g, \delta_c\}$. We partition all the possible rating matrices into subsets depending on $\delta$. Let $\mathcal{M}^{(\delta)}$ be the set of rating matrices subject to $\delta$.

**Problem of interest:** Our goal is to estimate a rating matrix $M \in \mathcal{M}^{(\delta)}$ given two types of information: (1) partial ratings $Y \in \{0, 1, *\}^{n \times m}$; (2) a graph, say social graph $\mathcal{G}$. Here $*$ indicates no observation, and we denote the set of observed entries of $Y$ by $\Omega$, that is $\Omega = \{(r, c) \in [n] \times [m] : Y_{rc} \neq *\}$. Below is a list of assumptions made for the analysis of the optimal sample complexity (Theorem 1) and theoretical guarantees of our proposed algorithm (Theorem 2), but not for the algorithm itself. We assume that each element of $Y$ is observed with probability $p \in [0, 1]$, independently from others, and its observation can possibly be flipped with probability $\theta \in [0, \frac{1}{2})$. Let social graph $\mathcal{G} = ([n], E)$ be an undirected graph, where $E$ denotes the set of edges, each capturing the social connection between two associated users. The set $[n]$ of vertices is partitioned into two disjoint clusters, each being further partitioned into three disjoint groups. We assume that the graph follows the hierarchical stochastic block model (HSBM) [51,59] with three types of edge probabilities: (i) $\alpha$ indicates an edge probability between two users in the *same group*; (ii) $\beta$ denotes the one w.r.t. two users of *different groups* yet within the *same cluster*; (iii) $\gamma$ is associated with two users of *different clusters*. We focus on realistic scenarios in which users within the same group (or cluster) are more likely to be connected as per the social homophily theory [19]: $\alpha \geq \beta \geq \gamma$.

**Performance metric:** Let $\psi$ be a rating matrix estimator that takes $(Y, \mathcal{G})$ as an input, yielding an estimate. As a performance metric, we consider the worst-case probability of error:

$$P_e^{(\delta)}(\psi) := \max_{M \in \mathcal{M}^{(\delta)}} \mathbb{P}\left[\psi(Y, \mathcal{G}) \neq M\right]. \tag{1}$$

Note that $\mathcal{M}^{(\delta)}$ is the set of ground-truth matrices $M$ subject to $\delta := \{\delta_g, \delta_c\}$. Since the error probability may vary depending on different choices of $M$ (i.e., some matrices may be harder to estimate), we employ a conventional minimax approach wherein the goal is to minimize the maximum error probability. We characterize the optimal sample complexity for reliable exact matrix recovery, concentrated around $nmp^\star$ in the limit of $n$ and $m$. Here $p^\star$ indicates the sharp threshold on the observation probability: (i) above which the error probability can be made arbitrarily close to 0 in the limit; (ii) under which $P_e^{(\delta)}(\psi) \nrightarrow 0$ no matter what and whatsoever.

## 3 Optimal sample complexity

We first present the optimal sample complexity characterized under the considered model. We find that an intuitive and insightful expression can be made via the quality of hierarchical social graph, which can be quantified by the following: (i) $I_g := (\sqrt{\alpha} - \sqrt{\beta})^2$ represents the capability of separating distinct *groups* within a cluster; (ii) $I_{c1} := (\sqrt{\alpha} - \sqrt{\gamma})^2$ and $I_{c2} := (\sqrt{\beta} - \sqrt{\gamma})^2$ capture the *clustering capabilities* of the social graph. Note that the larger the quantities, the easier

to do grouping/clustering. Our sample complexity result is formally stated below as a function of $(I_g, I_{c1}, I_{c2})$. As in [41], we make the same assumption on $m$ and $n$ that turns out to ease the proof via prominent large deviation theories: $m = \omega(\log n)$ and $\log m = o(n)$. This assumption is also practically relevant as it rules out highly asymmetric matrices.

**Theorem 1** (Information-theoretic limits). *Assume that $m = \omega(\log n)$ and $\log m = o(n)$. Let the item ratings be drawn from a finite field $\mathbb{F}_q$. Let $c$ and $g$ denote the number of clusters and groups, respectively. Within each cluster, let the set of $g$ rating vectors be spanned by any $r \leq g$ vectors in the same set. Define $p^\star$ as*

$$p^\star := \frac{1}{\left(\sqrt{1-\theta} - \sqrt{\frac{\theta}{q-1}}\right)^2} \max \left\{ \frac{gc}{g-r+1} \frac{\log m}{n}, \frac{\log n - \frac{n}{gc}I_g}{\delta_g m}, \frac{\log n - \frac{n}{gc}I_{c1} - \frac{(g-1)n}{gc}I_{c2}}{\delta_c m} \right\}. \quad (2)$$

*Fix $\epsilon > 0$. If $p \geq (1+\epsilon)p^\star$, then there exists a sequence of estimators $\psi$ satisfying $\lim_{n\to\infty} P_e^{(\delta)}(\psi) = 0$. Conversely, if $p \leq (1-\epsilon)p^\star$, then $\lim_{n\to\infty} P_e^{(\delta)}(\psi) \neq 0$ for any $\psi$. Setting $(c, g, r, q) = (2, 3, 2, 2)$, the bound in (1) reduces to*

$$p^\star = \frac{1}{(\sqrt{1-\theta} - \sqrt{\theta})^2} \max \left\{ \frac{3 \log m}{n}, \frac{\log n - \frac{1}{6}nI_g}{m\delta_g}, \frac{\log n - \frac{1}{6}nI_{c1} - \frac{1}{3}nI_{c2}}{m\delta_c} \right\}, \quad (3)$$

*which is the optimal sample complexity of the problem formulated in Section 2.*

*Proof.* We provide the proof sketch for $(c, g, r) = (2, 3, 2)$. We defer the complete proof for $(c, g, r) = (2, 3, 2)$ to the supplementary material. The extension to general $(c, g, r)$ is a natural generalization of the analysis for the parameters $(c, g, r) = (2, 3, 2)$. The achievability proof is based on maximum likelihood estimation (MLE). We first evaluate the likelihood for a given clustering/grouping of users and the corresponding rating matrix. We then show that if $p \geq (1+\epsilon)p^\star$, the likelihood is maximized only by the ground-truth rating matrix in the limit of $n$: $\lim_{n\to\infty} P_e^{(\delta)}(\psi_{\mathsf{ML}}) = 0$. For the converse (impossibility) proof, we first establish a lower bound on the error probability, and show that it is minimized when employing the maximum likelihood estimator. Next we prove that if $p$ is smaller than any of the three terms in the RHS of (3), then there exists another solution that yields a larger likelihood, compared to the ground-truth matrix. More precisely, if $p \leq \frac{(1-\epsilon)3\log m}{(\sqrt{1-\theta}-\sqrt{\theta})^2 n}$, we can find a grouping with the only distinction in two user-item pairs relative to the ground truth, yet yielding a larger likelihood. Similarly when $p \leq \frac{(1-\epsilon)(\log n - \frac{1}{6}nI_g)}{(\sqrt{1-\theta}-\sqrt{\theta})^2 m\delta_g}$, consider two users in the same cluster yet from distinct groups such that the hamming distance between their rating vectors is $m\delta_g$. We can then show that a grouping in which their rating vectors are swapped provides a larger likelihood. Similarly when $p \leq \frac{(1-\epsilon)(\log n - \frac{1}{6}nI_{c1} - \frac{1}{3}nI_{c2})}{(\sqrt{1-\theta}-\sqrt{\theta})^2 m\delta_c}$, we can swap the rating vectors of two users from different clusters with a hamming distance of $m\delta_c$, and get a greater likelihood.

The technical distinctions w.r.t. the prior works [41,42] are three folded: (i) the likelihood computation requires more involved combinatorial arguments due to the hierarchical structure; (ii) sophisticated upper/lower bounding techniques are developed in order to exploit the relational structure across different groups; (iii) delicate choices are made for two users to be swapped in the converse proof. □

We next present the second yet more practically-appealing contribution: Our proposed algorithm in Section 4 achieves the information-theoretic limits. The algorithm optimality is guaranteed for a certain yet wide range of scenarios in which graph information yields negligible clustering/grouping errors, formally stated below. We provide the proof outline in Section 4 throughout the description of the algorithm, leaving details in the supplementary material.

**Theorem 2** (Theoretical guarantees of the proposed algorithm). *Assume that $m = \omega(\log n)$, $\log m = o(n)$, $m = O(n)$, $I_{c2} > \frac{2\log n}{n}$ and $I_g > \omega(\frac{1}{n})$. Then, as long as the sample size is beyond the optimal sample complexity in Theorem 1 (i.e., $mnp > mnp^\star$), then the algorithm presented in Section 4 with $T = O(\log n)$ iterations ensures the worse-case error probability tends to 0 as $n \to \infty$. That is, the algorithm returns $\widehat{M}$ such that $\mathbb{P}[\widehat{M} = M] = 1 - o(1)$.*

Theorem 1 establishes the optimal sample complexity (the number of entries of the rating matrix to be observed) to be $mnp^\star$, where $p^\star$ is given in (3). The required sample complexity is a non-increasing

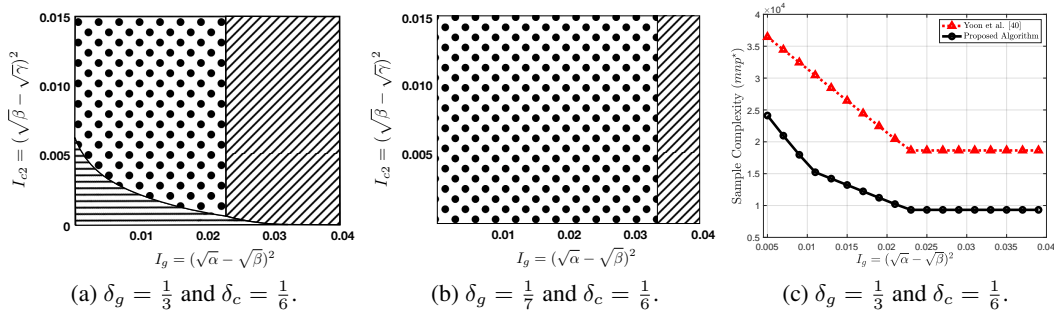

(a) $\delta_g = \frac{1}{3}$ and $\delta_c = \frac{1}{6}$.      (b) $\delta_g = \frac{1}{7}$ and $\delta_c = \frac{1}{6}$.      (c) $\delta_g = \frac{1}{3}$ and $\delta_c = \frac{1}{6}$.

Figure 1: Let $n = 1000$, $m = 500$ and $\theta = 0$. (a), (b) The different regimes of the optimal sample complexity reported in (3), where in (a) $\delta_c < \delta_g$ and in (b) $\delta_c > \delta_g$. Diagonal stripes, dots, and horizontal stripes refer to perfect clustering/grouping, grouping-limited, and clustering-limited regimes, respectively. (c) Comparison between the sample complexity reported in (3) for $\gamma = 0.01$ and $I_{c2} = 0.002$ and that of [42].

function of $\delta_g$ and $\delta_c$. This makes an intuitive sense because increasing $\delta_g$ (or $\delta_c$) yields more distinct rating vectors, thus ensuring easier grouping (or clustering). We emphasize three regimes depending on $(I_g, I_{c1}, I_{c2})$. The first refers to the so-called *perfect clustering/grouping regime* in which $(I_g, I_{c1}, I_{c2})$ are large enough, thereby activating the 1st term in the max function. The second is the *grouping-limited regime*, in which the quantity $I_g$ is not large enough so that the 2nd term becomes dominant. The last is the *clustering-limited regime* where the 3rd term is activated. A few observations are in order. For illustrative simplicity, we focus on the noiseless case, i.e., $\theta = 0$.

**Remark 1** (Perfect clustering/grouping regime). *The optimal sample complexity reads $3m \log m$. This result is interesting. A naive generalization of [41, 42] requires $4m \log m$, as we have four rating vectors $(v_1^A, v_2^A, v_1^B, v_2^B)$ to estimate and each requires $m \log m$ observations under our random sampling, due to the coupon-collecting effect. On the other hand, we exploit the relational structure across rating vectors of different group, reflected in $v_3^A = v_1^A \oplus v_2^A$ and $v_3^B = v_1^B \oplus v_2^B$; and we find this serves to estimate $(v_1^A, v_2^A, v_1^B, v_2^B)$ more efficiently, precisely by a factor of $\frac{4}{3}$ improvement, thus yielding $3m \log m$. This exploitation is reflected as novel technical contributions in the converse proof, as well as the achievability proofs of MLE and the proposed algorithm.* ∎

**Remark 2** (Grouping-limited regime). *We find that the sample complexity $\frac{n \log n - \frac{1}{6} n^2 I_g}{\delta_g}$ in this regime coincides with that of [42]. This implies that exploiting the relational structure across different groups does not help improving sample complexity when grouping information is not reliable.* ∎

**Remark 3** (Clustering-limited regime). *This is the most challenging scenario which has not been explored by any prior works. The challenge is actually reflected in the complicated sample complexity formula: $\frac{n \log n - \frac{1}{6} n^2 I_{c1} - \frac{1}{3} n^2 I_{c2}}{\delta_c}$. When $\beta = \gamma$, i.e., groups and clusters are not distinguishable, $I_g = I_{c1}$ and $I_{c2} = 0$. Therefore, in this case, it indeed reduces to a 6-group setting: $\frac{n \log n - \frac{1}{6} n^2 I_g}{\delta_c}$. The only distinction appears in the denominator. We read $\delta_c$ instead of $\delta_g$ due to different rating vectors across clusters and groups. When $I_{c2} \neq 0$, it reads the complicated formula, reflecting non-trivial technical contribution as well.* ∎

Fig. 1 depicts the different regimes of the optimal sample complexity as a function of $(I_g, I_{c2})$ for $n = 1000$, $m = 500$ and $\theta = 0$. In Fig. 1a, where $\delta_g = \frac{1}{3}$ and $\delta_c = \frac{1}{6}$, the region depicted by diagonal stripes corresponds to the perfect clustering/grouping regime. Here, $I_g$ and $I_{c2}$ are large, and graph information is rich enough to perfectly retrieve the clusters and groups. In this regime, the 1st term in (3) dominates. The region shown by dots corresponds to grouping-limited regime, where the 2nd term in (3) is dominant. In this regime, graph information suffices to exactly recover the clusters, but we need to rely on rating observation to exactly recover the groups. Finally, the 3rd term in (3) dominates in the region captured by horizontal stripes. This indicates the clustering-limited regimes, where neither clustering nor grouping is exact without the side information of the rating vectors. It is worth noting that in practically-relevant systems, where $\delta_c > \delta_g$ (for rating vectors of users in the same cluster are expected to be more similar compared to those in a different cluster), the third regime vanishes, as shown by Fig. 1b, where $\delta_g = \frac{1}{7}$ and $\delta_c = \frac{1}{6}$. It is straightforward to show that the third term in (3) is inactive whenever $\delta_c > \delta_g$. Fig. 1c compares the optimal sample

complexity between the one reported in (3), as a function of $I_g$, and that of [42]. The considered setting is $n = 1000$, $m = 500$, $\theta = 0$, $\delta_g = \frac{1}{3}$, $\delta_c = \frac{1}{6}$, $\gamma = 0.01$ and $I_{c2} = 0.002$. Note that [42] leverages neither the hierarchical structure of the graph, nor the linear dependency among the rating vectors. Thus, the problem formulated in Section 2 will be translated to a graph with six clusters with linearly independent rating vectors in the setting of [42]. Also, the minimum hamming distance for [42] is $\delta_c$. In Fig. 1c, we can see that the noticeable gain in the sample complexity of our result in the diagonal parts of the plot (the two regimes on the left side) is due to leveraging the hierarchical graph structure, while the improvement in the sample complexity in the flat part of the plot is a consequence of exploiting the linear dependency among the rating vectors within each cluster (See Remark 1).

## 4    Proposed Algorithm

We propose a computationally feasible matrix completion algorithm that achieves the optimal sample complexity characterized by Theorem 1. The proposed algorithm is motivated by a line of research on iterative algorithms that solve non-convex optimization problems [6, 58, 60–70]. The idea is to first find a good initial estimate, and then successively refine this estimate until the optimal solution is reached. This approach has been employed in several problems such as matrix completion [6, 60], community recovery [58, 61–63], rank aggregation [64], phase retrieval [65, 66], robust PCA [67], EM-algorithm [68], and rating estimation in crowdsourcing [69, 70]. In the following, we describe the proposed algorithm that consists of four phases to recover clusters, groups and rating vectors. Then, we discuss the computational complexity of the algorithm.

Recall that $Y \in \{0, +1, *\}^{n \times m}$. For the sake of tractable analysis, it is convenient to map $Y$ to $Z \in \{-1, 0, +1\}^{n \times m}$ where the mapping of the alphabet of $Y$ is as follows: $0 \longleftrightarrow +1$, $+1 \longleftrightarrow -1$ and $* \longleftrightarrow 0$. Under this mapping, the modulo-2 addition over $\{0, 1\}$ in $Y$ is represented by the multiplication of integers over $\{+1, -1\}$ in $Z$. Also, note that all recovery guarantees are asymptotic, i.e., they are characterized with high probability as $n \to \infty$. Throughout the design and analysis of the proposed algorithm, the number and size of clusters and groups are assumed to be known.

### 4.1    Algorithm Description

**Phase 1 (Exact Recovery of Clusters):** We use the community detection algorithm in [57] on $\mathcal{G}$ to *exactly* recover the two clusters $A$ and $B$. As proved in [57], the decomposition of the graph into two clusters is correct with high probability when $I_{c2} > \frac{2 \log n}{n}$.

**Phase 2 (Almost Exact Recovery of Groups):** The goal of Phase 2 is to decompose the set of users in cluster $A$ (cluster $B$) into three groups, namely $G_1^A, G_2^A, G_3^A$ (or $G_1^B, G_2^B, G_3^B$ for cluster $B$). It is worth noting that grouping at this stage is *almost exact*, and will be further refined in the next phases. To this end, we run a spectral clustering algorithm [58] on $A$ and $B$ separately. Let $\widehat{G}_i^x(0)$ denote the initial estimate of the $i^{\text{th}}$ group of cluster $x$ that is recovered by Phase 2 algorithm, for $i \in [3]$ and $x \in \{A, B\}$. It is shown that the groups within each cluster are recovered with a vanishing fraction of error if $I_g = \omega(1/n)$. It is worth mentioning that there are other clustering algorithms [62, 71–77] that can be employed for this phase. Examples include: spectral clustering [62, 71–74], semidefinite programming (SDP) [75], non-backtracking matrix spectrum [76], and belief propagation [77].

**Phase 3 (Exact Recovery of Rating Vectors):** We propose a novel algorithm that optimally recovers the rating vectors of the groups within each cluster. The algorithm is based on maximum likelihood (ML) decoding of users' ratings based on the partial and noisy observations. For this model, the ML decoding boils down to a counting rule: for each item, find the group with maximum gap between the number of observed zeros and ones, and set the rating entry of this group to $0$. The other two rating vectors are either both $0$ or both $1$ for this item, which will be determined based on the majority of the union of their observed entries. It turns out that the vector recovery is exact with probability $1 - o(1)$. This is one of the technical distinctions, relative to the prior works [41, 42] which employ the simple majority voting rule under non-hierarchical SBMs.

Define $\widehat{v}_i^x$ as the estimated rating vector of $v_i^x$, i.e., the output of Phase 3 algorithm. Let the $c^{\text{th}}$ element of the rating vector $v_i^x$ (or $\widehat{v}_i^x$) be denoted by $v_i^x(c)$ (or $\widehat{v}_i^x(c)$), for $i \in [3]$, $x \in \{A, B\}$ and $c \in [m]$. Let $Y_{r,c}$ be the entry of matrix $Y$ at row $r$ and column $c$, and $Z_{r,c}$ be its mapping to $\{+1, 0, -1\}$. The pseudocode of Phase 3 algorithm is given by Algorithm 1.

---
**Algorithm 1** Exact Recovery of Rating Vectors
---
1: **function** VECRCV $(n, m, Z, \{\widehat{G}_i^x(0) : i \in [3], x \in \{A, B\}\})$
2:      **for** $c \in [m]$ and $x \in \{A, B\}$ **do**
3:          **for** $i \in [3]$ **do** $\rho_{i,x}(c) \leftarrow \sum_{r \in \widehat{G}_i^x(0)} Z_{r,c}$
4:          $j \leftarrow \arg\max_{i \in [3]} \rho_{i,x}(c)$
5:          $\widehat{v}_j^x(c) \leftarrow 0$
6:          **if** $\sum_{i \in [3] \setminus \{j\}} \rho_{i,x}(c) \geq 0$ **then**
7:              **for** $i \in [3] \setminus \{j\}$ **do** $\widehat{v}_i^x(c) \leftarrow 0$
8:          **else**
9:              **for** $i \in [3] \setminus \{j\}$ **do** $\widehat{v}_i^x(c) \leftarrow 1$
10:     **return** $\{\widehat{v}_i^x : i \in [3], x \in \{A, B\}\}$
---

---
**Algorithm 2** Local Iterative Refinement of Groups (Set $flag = 0$)
---
1: **function** REFINE $(flag, n, m, T, Y, Z, \mathcal{G}, \{(\widehat{G}_i^x(0), \widehat{v}_i^x) : i \in [3], x \in \{A, B\}\})$
2:      $\widehat{\alpha} \leftarrow \frac{1}{6\binom{n/6}{2}} |\{(f,g) \in E : f, g \in G_i^x, x \in \{A, B\}, i \in [3]\}|$
3:      $\widehat{\beta} \leftarrow \frac{6}{n^2} |\{(f,g) \in E : f \in G_i^x, g \in G_j^x, x \in \{A, B\}, i \in [3], j \in [3] \setminus i\}|$
4:      $\widehat{\theta} \leftarrow |\{(r,c) \in \Omega : Y_{rc} \neq \widehat{v}_i^x(c), r \in \widehat{G}_i^x(0)\}|/|\Omega|$
5:      **for** $t \in [T]$ and $x \in \{A, B\}$ **do**
6:          **for** $i \in [3]$ **do** $\widehat{G}_i^x(t) \leftarrow \varnothing$
7:          **for** $r \leftarrow 1$ **to** $n$ **do**
8:              $j \leftarrow \arg\max_{i \in [3]} |\{c : Y_{r,c} = \widehat{v}_i^x(c)\}| \cdot \log\left(\frac{1-\widehat{\theta}}{\widehat{\theta}}\right) + e\left(\{r\}, \widehat{G}_i^x(t-1)\right) \cdot \log\left(\frac{(1-\widehat{\beta})\widehat{\alpha}}{(1-\widehat{\alpha})\widehat{\beta}}\right)$
9:              $\widehat{G}_j^x(t) \leftarrow \widehat{G}_j^x(t) \cup \{r\}$
10:         **if** $flag == 1$ **then**
11:             $\{\widehat{v}_i^x : i \in [3], x \in \{A, B\}\} \leftarrow$ VECRCV $(n, m, Z, \{\widehat{G}_i^x(t) : i \in [3], x \in \{A, B\}\})$
12:    **return** $\{\widehat{G}_i^x(T) : i \in [3], x \in \{A, B\}\}, \{\widehat{v}_i^x : i \in [3], x \in \{A, B\}\}$
---

**Phase 4 (Exact Recovery of Groups):** Finally, the goal is to *refine* the groups which are *almost recovered* in Phase 2, to obtain an *exact* grouping. To this end, we propose an iterative algorithm that locally refines the estimates on the user grouping within each cluster for $T$ iterations. Specifically, at each iteration, the affiliation of each user is updated to the group that yields the maximum local likelihood. This is determined based on (i) the number of edges between the user and the set of users which belong to that group, and (ii) the number of observed rating matrix entries of the user that coincide with the corresponding entries of the rating vector of that group. Algorithm 2 describes the pseudocode of Phase 4 algorithm. Note that we do not assume the knowledge of the model parameters $\alpha$, $\beta$ and $\theta$, and estimate them using $Y$ and $\mathcal{G}$, i.e., the proposed algorithm is parameter-free.

In order to prove the exact recovery of groups after running Algorithm 2, we need to show that the number of misclassified users in each cluster strictly decreases with each iteration of Algorithm 2. More specifically, assuming that the previous phases are executed successfully, if we start with $\eta n$ misclassified users within one cluster, for some small $\eta > 0$, then one can show that we end up with $\frac{\eta}{2} n$ misclassified users with high probability as $n \to \infty$ after one iteration of refinement. Hence, running the local refinement for $T = \frac{\log(\eta n)}{\log 2}$ within the groups of each cluster would suffice to converge to the ground truth assignments. The analysis of this phase follows the one in [42, Theorem 2] in which the problem of recovering $K$ communities of possibly different sizes is studied. By considering the case of three equal-sized communities, the guarantees of exact recovery of the groups within each cluster readily follows when $T = O(\log n)$.

**Remark 4.** *The iterative refinement in Algorithm 2 can be applied only on the groups (when $flag = 0$), or on the groups as well as the rating vectors (for $flag = 1$). Even though the former is sufficient for reliable estimation of the rating matrix, we show, through our simulation results in the following section, that the latter achieves a better performance for finite regimes of $n$ and $m$.* ∎

**Remark 5.** *The problem is formulated under the finite-field model only for the purpose of making an initial step towards a more generalized and realistic algorithm. Fortunately, as many of the*

*theory-inspiring works do, the theory process of characterizing the optimal sample complexity under this model could also shed insights into developing a universal algorithm that is applicable to a general problem setting rather than the specific problem setting considered for the theoretical analysis, as long as some slight algorithmic modifications are made. To demonstrate the universality of the algorithm, we consider a practical scenario in which ratings are real-valued (for which linear dependency between rating vectors is well-accepted) and observation noise is Gaussian. In this setting, the* detection *problem (under the current model) will be replaced by an* estimation *problem. Consequently, we update Algorithm 1 to incorporate an MLE of the rating vectors; and modify the local refinement criterion on Line 8 in Algorithm 2 to find the group that minimizes some properly-defined distance metric between the observed and estimated ratings such as Root Mean Squared Error (RMSE). In Section 5, we conduct experiments under the aforementioned setting, and show that our algorithm achieves superior performance over the state-of-the-art algorithms.* ∎

## 4.2 Computational Complexity

One of the crucial aspects of the proposed algorithm is its computational efficiency. Phase 1 can be done in polynomial time in the number of vertices [57,78]. Phase 2 can be done in $O(|E| \log n)$ using the power method [79]. Phase 3 requires a single pass over all entries of the observed matrix, which corresponds to $O(|\Omega|)$. Finally, in each iteration of Phase 4, the affiliation update of user $r \in [n]$ requires reading the entries of the $r^{\text{th}}$ row of $Y$ and the edges connected to user $r$, which amounts to $O(|\Omega| + |E|)$ for each of the $T$ iterations, assuming an appropriate data structure. Hence, the overall computational complexity reads $\text{poly}(n) + O(|\Omega| \log n)$.

**Remark 6.** *The complexity bottleneck is in Phase 1 (exact clustering), as it relies upon [57, 78], exhibiting* $\text{poly}(n)$ *runtime. This can be improved, without any performance degradation, by replacing the* exact *clustering in Phase 1 with* almost exact *clustering, yielding* $O(|E| \log n)$ *runtime [79]. In return, Phase 4 should be modified so that the local iterative refinement is applied on cluster affiliation, as well as group affiliation and rating vectors. As a result, the improved overall runtime reads* $O((|\Omega| + |E|) \log n)$. ∎

## 5 Experimental Results

We first conduct Monte Carlo experiments to corroborate Theorem 1. Let $\alpha = \widetilde{\alpha} \frac{\log n}{n}$, $\beta = \widetilde{\beta} \frac{\log n}{n}$, and $\gamma = \widetilde{\gamma} \frac{\log n}{n}$. We consider a setting where $\theta = 0.1, \widetilde{\beta} = 10, \widetilde{\gamma} = 0.5, \delta_g = \delta_c = 0.5$. The synthetic data is generated as per the model in Section 2. In Figs. 2a and 2b, we evaluate the performance of the proposed algorithm (with local iterative refinement of groups and rating vectors), and quantify the empirical success rate as a function of the normalized sample complexity, over $10^3$ randomly drawn realizations of rating vectors and hierarchical graphs. We vary $n$ and $m$, preserving the ratio $n/m = 3$. Fig. 2a depicts the case of $\widetilde{\alpha} = 40$ which corresponds to perfect clustering/grouping regime (Remark 1). On the other hand, Fig. 2b depicts the case of $\widetilde{\alpha} = 40$ which corresponds to grouping-limited regime (Remark 2). In both figures, we observe a phase transition[2] in the success rate at $p = p^\star$, and as we increase $n$ and $m$, the phase transition gets sharper. These figures corroborate Theorem 1 in different regimes when the graph side information is not scarce. Fig. 2c compares the performance of the proposed algorithm for $n = 3000$ and $m = 1000$ under two different strategies of local iterative refinement: (i) local refinement of groups only (set $flag = 0$ in Algorithm 2); and (ii) local refinement of both groups and rating vectors (set $flag = 1$ in Algorithm 2). It is clear that the second strategy outperforms the first in the finite regime of $n$ and $m$, which is consistent with Remark 4. Furthermore, the gap between the two versions shrinks as we gradually increase $\widetilde{\alpha}$ (i.e., as the quality of the graph gradually improves).

Next, similar to [41–44], the performance of the proposed algorithm is assessed on semi-real data (real graph but synthetic rating vectors). We consider a subgraph of the political blog network [80], which is shown to exhibit a hierarchical structure [50]. In particular, we consider a tall matrix setting of $n = 381$ and $m = 200$ in order to investigate the gain in sample complexity due to the graph side information. The selected subgraph consists of two clusters of political parties, each of which comprises three groups. The three groups of the first cluster consist of 98, 34 and 103 users, while the

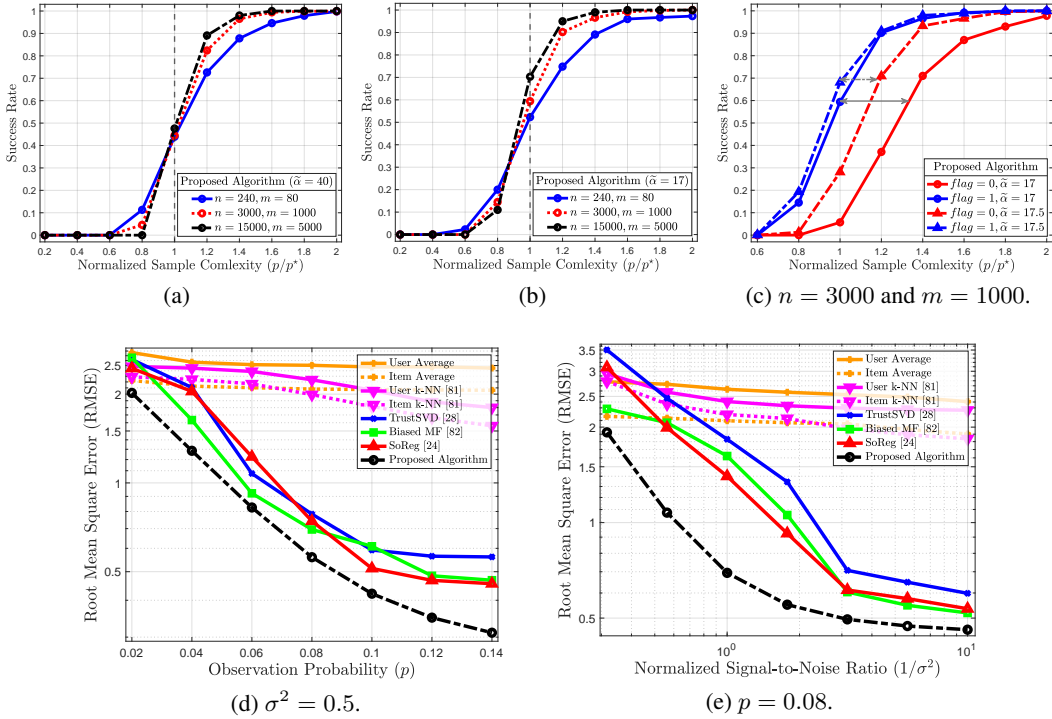

Figure 2: (a), (b) The success rate of the proposed algorithm as a function of $p/p^\star$ under different values of $n$, $m$ and $\widetilde{\alpha}$, where (a) corresponds to perfect clustering/grouping regime, and (b) corresponds to grouping-limited regime. (c) Performance of the proposed algorithm under two different local iterative refinement strategies. (d) Comparison of RMSE achieved by various recommendation algorithm on the Poliblog dataset [80] as a function of $p$. (e) Comparison of RMSE achieved by various recommendation algorithm on the Poliblog dataset as a function of $1/\sigma^2$.

Table 1: Runtimes of recommendation algorithms for the experiment setting of Fig. 2d and $p = 0.1$.

|  | User Average | Item Average | User k-NN | Item k-NN | TrustSVD | Biased MF | SoReg | Proposed Algorithm |
|---|---|---|---|---|---|---|---|---|
| Time (sec) | 0.021 | 0.025 | 0.299 | 0.311 | 0.482 | 0.266 | 0.328 | 0.055 |

three groups of the second cluster consist of 58, 68 and 20 users[3]. The corresponding rating vectors are generated such that the ratings are drawn from $[0, 10]$ (i.e., real numbers), and the observations are corrupted by a Gaussian noise with mean zero and a given variance $\sigma^2$. We use root mean square error (RMSE) as the evaluation metric, and assess the performance of the proposed algorithm against various recommendation algorithms, namely User Average, Item Average, User k-Nearest Neighbor (k-NN) [81], Item k-NN [81], TrustSVD [28], Biased Matrix Factorization (MF) [82], and Matrix Factorization with Social Regularization (SoReg) [24]. Note that [41, 42] are designed to work for rating matrices whose elements are drawn from a finite field, and hence they cannot be run under the practical scenario considered in this setting. In Fig. 2d, we compute RMSE as a function of $p$, for fixed $\sigma^2 = 0.5$. On the other hand, Fig. 2e depicts RMSE as a function of the normalized signal-to-noise ratio $1/\sigma^2$, for fixed $p = 0.08$. It is evident that the proposed algorithm achieves superior performance over the state-of-the-art algorithms for a wide range of observation probabilities and Gaussian noise variances, demonstrating its viability and efficiency in practical scenarios.

Finally, Table 1 demonstrates the computational efficiency of the proposed algorithm, and reports the runtimes of recommendation algorithms for the experiment setting of Fig. 2d and $p = 0.1$. The runtimes are averaged over 20 trials. The proposed algorithm achieves a faster runtime than all other algorithms except for User Average and Item Average. However, as shown in Fig. 2d, the performance of these faster algorithms, in terms of RMSE, is inferior to the majority of other algorithms.

## Broader Impact

We emphasize two positive impacts of our work. First, it serves to enhance the performance of *personalized* recommender systems (one of the most influential commercial applications) with the aid of social graph which is often available in a variety of applications. Second, it achieves *fairness* among all users by providing high quality recommendations even to new users who have not rated any items before. One negative consequence of this work is w.r.t. the *privacy* of users. User privacy may not be preserved in the process of exploiting *indirect* information posed in social graphs, even though direct information, such as user profiles, is protected.

## Acknowledgments and Disclosure of Funding

The work of A. Elmahdy and S. Mohajer is supported in part by the National Science Foundation under Grants CCF-1617884 and CCF-1749981. The work of J. Ahn and C. Suh is supported by the National Research Foundation of Korea (NRF) grant funded by the Korea government (MSIP) (No.2018R1A1A1A05022889).

## Footnotes

[2]The transition is ideally a step function at $p = p^\star$ as $n$ and $m$ tend to infinity.

[3]We refer to the supplementary material for a visualization of the selected subgraph of the political blog network using t-SNE algorithm.

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
