[Supplementary Material]

# Supplementary Material
# Matrix Completion with Hierarchical Graph Side Information

**Adel Elmahdy**[*]
ECE, University of Minnesota
adel@umn.edu

**Junhyung Ahn**[*]
EE, KAIST
tonyahn96@kaist.ac.kr

**Changho Suh**
EE, KAIST
chsuh@kaist.ac.kr

**Soheil Mohajer**
ECE, University of Minnesota
soheil@umn.edu

## Contents

---

[*]Equal Contribution

# 1 List of Underlying Assumptions

The proofs of Theorem 1 and Theorem 2 rely on a number of assumptions on the model parameters $(n, m, p, \theta, \alpha, \beta, \gamma)$. We enumerate them before proceeding with the formal proofs in the following sections.

- $n$ and $m$ tend to $\infty$.
- $m = \omega(\log n)$ and $\log m = o(n)$. These assumptions rule out extremely tall or wide matrices, respectively, so that we can resort to large deviation theories in the proofs.
- $m = O(n)$. This is a sufficient condition for reliable estimation of $(\alpha, \beta, \theta)$ for the proposed computationally-efficient algorithm. If these parameters are known a priori, this assumption can be disregarded.
- $\theta = \Theta(1)$.
- $\alpha \geq \beta \geq \gamma$. This assumption reflects realistic scenarios in which users within the same group (or cluster) are more likely to be connected as per the social homophily theory [1].
- $\alpha, \beta, \gamma = \Theta\left(\frac{\log n}{n}\right)$.

# 2 Proof of Theorem 1

**Theorem 1** (Information-theoretic limits). *Assume that $m = \omega(\log n)$ and $\log m = o(n)$. Let*

$$p^{\star} := \frac{1}{(\sqrt{1-\theta} - \sqrt{\theta})^2} \max\left\{ \frac{3\log m}{n}, \frac{\log n - \frac{1}{6}nI_g}{m\delta_g}, \frac{\log n - \frac{1}{6}nI_{c1} - \frac{1}{3}nI_{c2}}{m\delta_c} \right\}. \tag{1}$$

*Fix $\epsilon > 0$. If $p \geq (1+\epsilon)p^{\star}$, then there exists a sequence of estimators $\psi$ satisfying $\lim_{n\to\infty} P_e^{(\delta)}(\psi) = 0$. Conversely, if $p \leq (1-\epsilon)p^{\star}$, then $\lim_{n\to\infty} P_e^{(\delta)}(\psi) \neq 0$ for any $\psi$.*

## 2.1 Achievability proof

Let $\psi_{\mathrm{ML}}$ be the maximum likelihood estimator. Fix $\epsilon > 0$. Consider the sufficient conditions claimed in Theorem 1:

$$p \geq \frac{(1+\epsilon)3\log m}{(\sqrt{1-\theta} - \sqrt{\theta})^2 n}, \tag{2}$$

$$p \geq \frac{(1+\epsilon)(\log n - \frac{1}{6}nI_g)}{(\sqrt{1-\theta} - \sqrt{\theta})^2 m\delta_g}, \tag{3}$$

$$p \geq \frac{(1+\epsilon)(\log n - \frac{1}{6}nI_{c1} - \frac{1}{3}nI_{c2})}{(\sqrt{1-\theta} - \sqrt{\theta})^2 m\delta_c}. \tag{4}$$

For notational simplicity, let us define $I_r := p(\sqrt{1-\theta} - \sqrt{\theta})^2$. Then, the above conditions can be rewritten as:

$$\frac{1}{3}nI_r \geq (1+\epsilon)\log m, \tag{5}$$

$$m\delta_g I_r + \frac{1}{6}nI_g \geq (1+\epsilon)\log n, \tag{6}$$

$$m\delta_c I_r + \frac{1}{6}nI_{c1} + \frac{1}{3}nI_{c2} \geq (1+\epsilon)\log n. \tag{7}$$

In what follows, we will show that the probability of error when applying $\psi_{\mathrm{ML}}$ tends to zero if all of the above conditions are satisfied.

Recall that each cluster consists of three groups and the rating vectors of the three groups respect some dependency relationship, reflected in $v_3^A = v_1^A \oplus v_2^A$ and $v_3^B = v_1^B \oplus v_2^B$. Here, $v_i^x$ denotes the rating vector of the $i^{\text{th}}$ group in cluster $x$ where $i \in \{1,2,3\}$ and $x \in \{A, B\}$. This then motivates us to assume that without loss of generality, the ground-truth rating matrix, say $M_0 \in \mathcal{M}^{(\delta)}$, reads:

$$M_0 := \begin{bmatrix} \mathbf{1}_{\frac{n}{6} \times \tau_{000}m} & \mathbf{1}_{\frac{n}{6} \times \tau_{001}m} & \mathbf{1}_{\frac{n}{6} \times \tau_{010}m} & \mathbf{1}_{\frac{n}{6} \times \tau_{011}m} & \mathbf{1}_{\frac{n}{6} \times \tau_{100}m} & \mathbf{1}_{\frac{n}{6} \times \tau_{101}m} & \mathbf{1}_{\frac{n}{6} \times \tau_{110}m} & \mathbf{1}_{\frac{n}{6} \times \tau_{111}m} \\ \mathbf{0}_{\frac{n}{6} \times \tau_{000}m} & \mathbf{0}_{\frac{n}{6} \times \tau_{001}m} & \mathbf{0}_{\frac{n}{6} \times \tau_{010}m} & \mathbf{0}_{\frac{n}{6} \times \tau_{011}m} & \mathbf{1}_{\frac{n}{6} \times \tau_{100}m} & \mathbf{1}_{\frac{n}{6} \times \tau_{101}m} & \mathbf{1}_{\frac{n}{6} \times \tau_{110}m} & \mathbf{1}_{\frac{n}{6} \times \tau_{111}m} \\ \mathbf{1}_{\frac{n}{6} \times \tau_{000}m} & \mathbf{1}_{\frac{n}{6} \times \tau_{001}m} & \mathbf{1}_{\frac{n}{6} \times \tau_{010}m} & \mathbf{1}_{\frac{n}{6} \times \tau_{011}m} & \mathbf{0}_{\frac{n}{6} \times \tau_{100}m} & \mathbf{0}_{\frac{n}{6} \times \tau_{101}m} & \mathbf{0}_{\frac{n}{6} \times \tau_{110}m} & \mathbf{0}_{\frac{n}{6} \times \tau_{111}m} \\ \mathbf{0}_{\frac{n}{6} \times \tau_{000}m} & \mathbf{0}_{\frac{n}{6} \times \tau_{001}m} & \mathbf{1}_{\frac{n}{6} \times \tau_{010}m} & \mathbf{1}_{\frac{n}{6} \times \tau_{011}m} & \mathbf{0}_{\frac{n}{6} \times \tau_{100}m} & \mathbf{0}_{\frac{n}{6} \times \tau_{101}m} & \mathbf{1}_{\frac{n}{6} \times \tau_{110}m} & \mathbf{1}_{\frac{n}{6} \times \tau_{111}m} \\ \mathbf{0}_{\frac{n}{6} \times \tau_{000}m} & \mathbf{1}_{\frac{n}{6} \times \tau_{001}m} & \mathbf{0}_{\frac{n}{6} \times \tau_{010}m} & \mathbf{1}_{\frac{n}{6} \times \tau_{011}m} & \mathbf{0}_{\frac{n}{6} \times \tau_{100}m} & \mathbf{1}_{\frac{n}{6} \times \tau_{101}m} & \mathbf{0}_{\frac{n}{6} \times \tau_{110}m} & \mathbf{1}_{\frac{n}{6} \times \tau_{111}m} \\ \mathbf{0}_{\frac{n}{6} \times \tau_{000}m} & \mathbf{1}_{\frac{n}{6} \times \tau_{001}m} & \mathbf{1}_{\frac{n}{6} \times \tau_{010}m} & \mathbf{0}_{\frac{n}{6} \times \tau_{011}m} & \mathbf{0}_{\frac{n}{6} \times \tau_{100}m} & \mathbf{1}_{\frac{n}{6} \times \tau_{101}m} & \mathbf{1}_{\frac{n}{6} \times \tau_{110}m} & \mathbf{0}_{\frac{n}{6} \times \tau_{111}m} \end{bmatrix} \tag{8}$$

where $0 < \tau_\ell < 1$ for $\ell \in \{0,1\}^3$, and $\sum_{\ell \in \{0,1\}^3} \tau_\ell = 1$. Here, we divide the columns of $M_0$ into eight sections $\mathcal{T}_\ell$ where

$$\mathcal{T}_{b_1 b_2 b_3} = \left\{ c \in [m] : \text{column } c \text{ of } M_0 = \begin{bmatrix} \mathbf{1}_{\frac{n}{6}} & b_1 \mathbf{1}_{\frac{n}{6}} & (1 \oplus b_1)\mathbf{1}_{\frac{n}{6}} & b_2 \mathbf{1}_{\frac{n}{6}} & b_3 \mathbf{1}_{\frac{n}{6}} & (b_2 \oplus b_3)\mathbf{1}_{\frac{n}{6}} \end{bmatrix}^{\mathsf{T}} \right\},$$

for $b_1, b_2, b_3 \in \{0,1\}^3$, and we have $\tau_\ell = |\mathcal{T}_\ell|/m$. Accordingly, each row $v_i^x$ is further partitioned $\{v_i^x(\ell) : \ell \in \{0,1\}^3\}$.

By symmetry, $\mathbb{P}[\psi_{\mathrm{ML}}(Y, \mathcal{G}) \neq M]$ is the same for all $M$'s as long as the considered matrix respects the $\delta$-constraint, i.e., belongs to the class of $\mathcal{M}^{(\delta)}$ where $\delta := \{\delta_c, \delta_g\}$. Hence,

$$P_e^{(\delta)}(\psi_{\mathrm{ML}}) := \max_{M \in \mathcal{M}^{(\delta)}} \mathbb{P}[\psi_{\mathrm{ML}}(Y, \mathcal{G}) \neq M] = \mathbb{P}[\psi_{\mathrm{ML}}(Y, \mathcal{G}) \neq M_0]. \tag{9}$$

By applying the union bound together with the definition of MLE, we then obtain

$$P_e^{(\delta)}(\psi_{\mathrm{ML}}) \leq \sum_{X \neq M_0} \mathbb{P}\left[\mathsf{L}(M_0) \geq \mathsf{L}(X)\right] \tag{10}$$

where $\mathsf{L}(X)$ denotes the negative log-likelihood of a candidate matrix $X$. It turns out that an interested error event $\{\mathsf{L}(M_0) \geq \mathsf{L}(X)\}$ depends solely on two key parameters which dictate the relationship between $X$ and $M_0 \in \mathcal{M}^{(\delta)}$. Let us first introduce some notations relevant to the two parameters. Let $\{v_i^x : x \in \{A, B\}, i \in [3]\}$ be the rating vectors w.r.t. $M_0$. Let $\{u_i^x : x \in \{A, B\}, i \in [3]\}$ be the counterparts w.r.t. $X$. The first key parameter, which we denote by $k_{ij}^{xy}$, indicates the number of users in group $i$ of cluster $x$ whose rating vector $v_i^x$'s are swapped with the rating vectors $u_j^y$'s of users in group $j$ of cluster $y$. The second key parameter, which we denote by $d_i^x(\ell)$, is the hamming distance between $v_i^x(\ell)$ and $u_i^x(\ell)$: $d_{\mathrm{H}}\left(v_i^x(\ell), u_i^x(\ell)\right)$ where $v_i^x(\ell)$ denotes part of $v_i^x$ concerning column block $\ell \in \{0, 1\}^3$ (similarly for $u_i^x(\ell)$).

We also find that the following constraint w.r.t. $k_{ij}^{xy}$'s plays a role in deriving some useful bounds in Lemma 3 (to be stated later):

$$\sum_{y \in \{A, B\}} \sum_{j \in [3]} k_{ij}^{xy} \leq \frac{5}{36}n. \tag{11}$$

Lemma 1 (to be stated shortly) shows that the constraint comes without losing generality, i.e., the constraint does not prevent the representation of all of the possible matrices. To figure out what this means in detail, we first partition $\mathcal{M}^{(\delta)}$ into numerous matrix classes: $\mathcal{M}^{(\delta)} = \bigcup_T \mathcal{X}(T)$. Here each class, which we denote by $\mathcal{X}(T)$, is characterized by a tuple $T$ that concerns the two key parameters:

$$T = \left(\left\{k_{ij}^{xy}\right\}_{i,j \in [3],\, x,y \in \{A,B\}}, \left\{d_i^x(\ell)\right\}_{i \in [3],\, \ell \in \{0,1\}^3,\, x \in \{A,B\}}\right). \tag{12}$$

Here, the matrix class $\mathcal{X}(T)$ denotes the set of all rating matrices subject to $T$. Let $\mathcal{T}^{(\delta)}$ be the set of such tuples $T$'s that also satisfy

$$\sum_{y \in \{A, B\}} \sum_{j \in [3]} k_{ij}^{xy} \leq \frac{5}{36}n, \quad 0 \leq d_i^x(\ell) \leq m\tau_\ell. \tag{13}$$

**Lemma 1.** *Consider $X \in \mathcal{M}^{(\delta)}$. Then, there exists $T \in \mathcal{T}^{(\delta)}$ such that $X \in \mathcal{X}(T)$. This implies that $\mathcal{M}^{(\delta)} = \mathcal{T}^{(\delta)}$, i.e., the constraint (13) made in $\mathcal{T}^{(\delta)}$ does not lose any generality in matrix representation.*

We refer to Appendix A.1 for the proof of Lemma 1.

Using the introduced set $\mathcal{T}^{(\delta)}$ and the tuple $T$, we can then rewrite the RHS of (10) as:

$$\sum_{X \neq M_0} \mathbb{P}\left[\mathsf{L}(M_0) \geq \mathsf{L}(X)\right] = \sum_{T \in \mathcal{T}^{(\delta)}} \sum_{X \in \mathcal{X}(T)} \mathbb{P}\left[\mathsf{L}(M_0) \geq \mathsf{L}(X)\right]. \tag{14}$$

Lemma 2 stated below provides an upper bound on $\mathbb{P}\left[\mathsf{L}(M_0) \geq \mathsf{L}(X)\right]$ for $X \in \mathcal{X}(T)$ and $T \in \mathcal{T}^{(\delta)}$.

**Lemma 2.** *Let $B_i^{(p)} \overset{\mathrm{i.i.d}}{\sim} \mathsf{Bern}(p)$ $B_i^{(\theta)} \overset{\mathrm{i.i.d}}{\sim} \mathsf{Bern}(\theta)$, $B_i^{(\alpha)} \overset{\mathrm{i.i.d}}{\sim} \mathsf{Bern}(\alpha)$, $B_i^{(\beta)} \overset{\mathrm{i.i.d}}{\sim} \mathsf{Bern}(\beta)$ and $B_i^{(\gamma)} \overset{\mathrm{i.i.d}}{\sim} \mathsf{Bern}(\gamma)$. Let*

$$\begin{aligned} B \coloneqq\ & c_1 \sum_{i=1}^{n_1} B_i^{(p)}\left(2B_i^{(\theta)} - 1\right) + c_2 \sum_{j=n_1+1}^{n_1+n_2} \left(B_j^{(\beta)} - B_k^{(\alpha)}\right) \\ & + c_3 \sum_{k=n_1+n_2+1}^{n_1+n_2+n_3} \left(B_k^{(\gamma)} - B_k^{(\alpha)}\right) + c_4 \sum_{\ell=n_1+n_2+n_3+1}^{n_1+n_2+n_3+n_4} \left(B_\ell^{(\gamma)} - B_\ell^{(\beta)}\right). \end{aligned} \tag{15}$$

*where $n_1 \coloneqq \Lambda(M_0, X)$, $n_2 \coloneqq \eta_T^{\alpha \to \beta}$, $n_3 \coloneqq \eta_T^{\alpha \to \gamma}$ and $n_4 \coloneqq \eta_T^{\beta \to \gamma}$. Here, $\Lambda(M_0, X)$ indicates the number of distinct entries between $M_0$ and $X$. Moreover, $\eta_T^{x \to y}$ denotes the number of pairs of users who are originally connected with $x$-type edges (in light of $M_0$), but misclassified as $y$-type edges (in*

*view of X) where $x, y \in \{\alpha, \beta, \gamma\}$; see Appendix A.2 for its mathematical definition. Assume that $\alpha, \beta, \gamma, p = o(1)$. Then, for $X \in \mathcal{X}(T)$ and $T \in \mathcal{T}^{(\delta)}$, we have*

$$\mathbb{P}\left[\mathsf{L}\left(M_0\right) \geq \mathsf{L}(X)\right] = \mathbb{P}\left[B \geq 0\right] \tag{16}$$

$$\leq \exp\left(-\left(1 + o(1)\right)\left(n_1 I_r + n_2 I_g + n_3 I_{c1} + n_4 I_{c2}\right)\right). \tag{17}$$

We refer to Appendix A.2 for the proof of Lemma 2.

Applying (17) to the RHS of (14), we then get

$$\sum_{T \in \mathcal{T}^{(\delta)}} \sum_{X \in \mathcal{X}(T)} \mathbb{P}\left[\mathsf{L}(M_0) \geq \mathsf{L}(X)\right] \leq \sum_{T \in \mathcal{T}^{(\delta)}} |\mathcal{X}(T)| \exp\left(-(1 + o(1))K^T I\right), \tag{18}$$

where $K := [n_1, n_2, n_3, n_4]^T$ and $I := [I_r, I_g, I_{c1}, I_{c2}]^T$. We find that partitioning $\mathcal{T}^{(\delta)}$ further into $\mathcal{T}_1^{(\delta)}$ and the rest $\mathcal{T}^{(\delta)} \setminus \mathcal{T}_1^{(\delta)}$ serves to ease the proof:

$$\mathcal{T}_1^{(\delta)} := \left\{ \left(\{k_{ij}^{xy}\}_{i,j \in [3], x, y \in \{A, B\}}, \{d_i^x(\ell)\}_{i \in [3], \ell \in \{0,1\}^3, x \in \{A, B\}}\right) : k_{ij}^{xy} \leq \frac{\tau}{5}n, \, d_i^x(\ell) \leq \tau m \right\},$$

where $\tau$ is some constant that lies in between 0 and $\tau_\ell$. Using this further split, we can then rewrite the RHS of (18) as

$$\sum_{T \in \mathcal{T}^{(\delta)} \setminus \mathcal{T}_1^{(\delta)}} |\mathcal{X}(T)| \exp\left(-(1 + o(1))K^T I\right) + \sum_{T \in \mathcal{T}_1^{(\delta)}} |\mathcal{X}(T)| \exp\left(-(1 + o(1))K^T I\right). \tag{19}$$

This together with Lemmas 3 and 4 (stated below) yields

$$\sum_{T \in \mathcal{T}^{(\delta)}} \sum_{X \in \mathcal{X}(T)} \mathbb{P}\left[\mathsf{L}(M_0) \geq \mathsf{L}(X)\right] \longrightarrow 0 \quad \text{as } n, m \to \infty. \tag{20}$$

Applying this to (14) and (10), we conclude that $P_e^{(\delta)}(\psi_{\mathrm{ML}}) \to 0$ as $n$ and $m$ tend to infinity. This completes the achievability proof of Theorem 1. $\qquad\square$

**Lemma 3.** *The first term in* (19) *is upper-bounded by*

$$\sum_{T \in \mathcal{T}^{(\delta)} \setminus \mathcal{T}_1^{(\delta)}} |\mathcal{X}(T)| \exp\left(-(1 + o(1))K^T I\right) \leq 6^n 2^{6m} e^{-\Omega(nm)I_r} \leq \left(\frac{6}{n}\right)^n \left(\frac{2^6}{m}\right)^m. \tag{21}$$

*We see that the RHS of* (21) *tends to 0 as $n$ and $m$ go to infinity, thus leading the LHS to converge to 0 in the limit.*

We refer to Appendix A.3 for the proof of Lemma 3.

**Lemma 4.** *The second term in* (19) *is upper-bounded by*

$$\sum_{T \in \mathcal{T}_1^{(\delta)}} |\mathcal{X}(T)| \exp\left(-(1 + o(1))K^T I\right) \leq \sum_{T \in \mathcal{T}_1^{(\delta)}} \exp\left(-\frac{\epsilon}{4}d_t \log m - \frac{\epsilon}{2}(k_g + k_c) \log n\right), \tag{22}$$

*where*

$$k_g := \sum_{x \in \{A, B\}} \sum_{i, j \in [3]} k_{ij}^{xx},$$

$$k_c := \sum_{x \in \{A, B\}} \sum_{y \in \{A, B\} \setminus \{x\}} \sum_{i, j \in [3]} k_{ij}^{xy},$$

$$d_t := \sum_{x \in \{A, B\}} \sum_{\ell \in \{0,1\}^3} d_1^x(\ell) + d_2^x(\ell) + d_3^x(\ell).$$

*Note that the RHS of* (22) *converges to 0 as $n$ and $m$ tend to infinity.*

We refer to Appendix A.4 for the proof of Lemma 4.

**Remark 1** (Technical distinction). *One technical distinction relative to the previous works [2, 3] arises from the fact that in our setting, the hamming distances $(d_1^x(\ell), d_2^x(\ell), d_3^x(\ell))$ defined w.r.t. different groups yet within the same cluster are intimately related. Note that the rating vectors of $X \in \mathcal{M}^{(\delta)}$ are linearly dependent: $u_3^x = u_1^x \oplus u_2^x$ for $x \in \{A, B\}$. To carefully compute $d_3^x(\ell)$ as a function of $d_1^x(\ell)$ and $d_2^x(\ell)$, we introduce another quantity that represents the number of elements where $u_1^x$ and $v_1^x$ differ in column block $\ell$, which we denote by $I_\ell$, (also for $u_2^x$, $v_2^x$):*

$$d_{ov}^x(\ell) := |\{c \in I_\ell : v_1^x(c) \neq u_1^x(c), v_2^x(c) \neq u_2^x(c)\}|. \tag{23}$$

*By the dependency structure,*

$$d_3^x(\ell) = (d_1^x(\ell) - d_{ov}^x(\ell)) + (d_2^x(\ell) - d_{ov}^x(\ell)) = d_1^x(\ell) + d_2^x(\ell) - 2d_{ov}^x(\ell). \tag{24}$$

*This distinction affects all the detailed derivations through the achievability proof.* ∎

## 2.2 Converse Proof

Define $I_r := p(\sqrt{1-\theta} - \sqrt{\theta})^2$. The goal of the converse proof is to show that $P_e^{(\tau)}(\psi) \nrightarrow 0$ as $n \to \infty$ for any set of feasible rating matrices $\mathcal{M}^{(\delta)}$ and estimator $\psi$, if at least one of the following conditions is satisfied:

$$\frac{1}{3}nI_r \leq (1-\epsilon)\log m, \qquad \textbf{(Perfect clustering/grouping regime)} \tag{25}$$

$$\delta_g m I_r + \frac{1}{6}nI_g \leq (1-\epsilon)\log n, \qquad \textbf{(Grouping-limited regime)} \tag{26}$$

$$\delta_c m I_r + \frac{1}{6}nI_{c1} + \frac{1}{3}nI_{c2} \leq (1-\epsilon)\log n, \quad \textbf{(Clustering-limited regime)} \tag{27}$$

We first seek a lower bound on the infimum of the worst-case probability of error over all estimators. Let $\mathbf{M}$ be a random variable that denotes the hidden rating matrix (to be estimated) and is uniformly drawn from $\mathcal{M}^{(\delta)}$. Denote the *success* event of estimation of rating matrix by $S$, which is given by

$$S := \bigcap_{\substack{X \in \mathcal{M}^{(\delta)} \\ X \neq M_0}} (\mathsf{L}(X) > \mathsf{L}(M_0)). \tag{28}$$

From the definition of worst-case probability of error in (9), we obtain

$$\inf_\psi P_e^{(\tau)}(\psi) = \inf_\psi \max_{M \in \mathcal{M}^{(\delta)}} \mathbb{P}\left[\psi(Y^\Omega, G) \neq M\right]$$

$$\geq \inf_\psi \max_{M \in \mathcal{M}^{(\delta)}} \mathbb{P}\left[\psi(Y^\Omega, G) \neq M, \mathbf{M} = M\right]$$

$$= \inf_\psi \max_{M \in \mathcal{M}^{(\delta)}} \mathbb{P}\left[\psi(Y^\Omega, G) \neq M \mid \mathbf{M} = M\right] \tag{29}$$

$$= \inf_\psi \max_{M \in \mathcal{M}^{(\delta)}} \sum_{X \neq M} \mathbb{P}\left[\psi(Y^\Omega, G) = X \mid \mathbf{M} = M\right]$$

$$\geq \inf_\psi \max_{M \in \mathcal{M}^{(\delta)}} \sum_{\substack{X \in \mathcal{M}^{(\delta)} \\ X \neq M}} \mathbb{P}\left[\psi(Y^\Omega, G) = X \mid \mathbf{M} = M\right]$$

$$= \max_{M \in \mathcal{M}^{(\delta)}} \sum_{\substack{X \in \mathcal{M}^{(\delta)} \\ X \neq M}} \mathbb{P}\left[\psi_{\mathrm{ML}}(Y^\Omega, G) = X \mid \mathbf{M} = M\right] \tag{30}$$

$$\geq \sum_{\substack{X \in \mathcal{M}^{(\delta)} \\ X \neq M_0}} \mathbb{P}\left[\psi_{\mathrm{ML}}(Y^\Omega, G) = X \mid \mathbf{M} = M_0\right] \tag{31}$$

$$= \sum_{\substack{X \in \mathcal{M}^{(\delta)} \\ X \neq M_0}} \mathbb{P}\left[\mathsf{L}(X) \leq \mathsf{L}(M_0)\right] \tag{32}$$

$$\geq \mathbb{P}\left[\bigcup_{\substack{X \in \mathcal{M}^{(\delta)} \\ X \neq M_0}} (\mathsf{L}(X) \leq \mathsf{L}(M_0))\right] \tag{33}$$

$$= \mathbb{P}\left[S^c\right] \tag{34}$$

where (29) follows because $\mathbf{M}$ is uniformly distributed; (30) follows due to the fact that the maximum likelihood estimator is optimal under uniform prior; (31) follows since $M_0 \in \mathcal{M}^{(\delta)}$; (32) follows by the definition of negative log-likelihood in (78); (33) follows from union bound; and finally (34) follows from (28). Therefore, in order to show that $\lim_{n\to\infty} \inf_\psi P_e^{(\tau)}(\psi) \neq 0$, it suffices to show that $\lim_{n\to\infty} \mathbb{P}\left[S\right] = 0$.

Next, we show that $\lim_{n\to\infty} \mathbb{P}\left[S\right] = 0$ under each of the three conditions stated in (25), (26), (27), respectively. Before delving into the convergence proof, we present the following key lemma that is essential for developing the convergence analysis. In this lemma, we use $B^{(\mu)}$ to refer to a Bernoulli random variable with (fixed or asymptotic) parameter $\mu \in [0, 1]$, that is, $\mathbb{P}[B^{(\mu)} = 1] = 1 - \mathbb{P}[B^{(\mu)} = 0] = \mu$.

**Lemma 5.** *Assume that* $\alpha, \beta, \gamma, p = \Theta\left(\frac{\log n}{n}\right)$ *and* $\theta \in [0, 1]$ *is a constant. For positive integers* $n_1, n_2, n_3, n_4$ *satisfying* $\max\left\{pn_1, \sqrt{\alpha\beta}n_2, \sqrt{\alpha\gamma}n_3, \sqrt{\beta\gamma}n_4\right\} = \omega(1)$, *consider the sets of independent Bernoulli random variables* $\{B_i^{(p)} : i \in [n_1]\}$, $\{B_i^{(\theta)} : i \in [n_1]\}$, $\{B_i^{(\alpha)} : i \in [n_1 + 1 : n_3]\}$, $\{B_i^{(\beta)} : i \in [n_1 + 1 : n_2] \cup [n_1 + n_2 + n_3 + 1 : n_1 + n_2 + n_3 + n_4]\}$, *and* $\{B_i^{(\gamma)} : i \in [n_1 + n_2 + 1 : n_1 + n_2 + n_3 + n_4]\}$. *Define*

$$
B(n_1, n_2, n_3, n_4) := \sum_{i=1}^{n_1} \log\left(\frac{1-\theta}{\theta}\right) B_i^{(p)}\left(2B_i^{(\theta)} - 1\right) + \sum_{j=n_1+1}^{n_1+n_2} \log\left(\frac{(1-\beta)\alpha}{(1-\alpha)\beta}\right)\left(B_j^{(\beta)} - B_j^{(\alpha)}\right)
$$

$$
+ \sum_{k=n_1+n_2+1}^{n_1+n_2+n_3} \log\left(\frac{(1-\gamma)\alpha}{(1-\alpha)\gamma}\right)\left(B_k^{(\gamma)} - B_k^{(\alpha)}\right)
$$

$$
+ \sum_{\ell=n_1+n_2+n_3+1}^{n_1+n_2+n_3+n_4} \log\left(\frac{(1-\gamma)\beta}{(1-\beta)\gamma}\right)\left(B_\ell^{(\gamma)} - B_\ell^{(\beta)}\right).
$$

*Then, the probability that* $B(n_1, n_2, n_3, n_4)$ *being non-negative can be lower bounded by*

$$
\mathbb{P}\left[B(n_1, n_2, n_3, n_4) \geq 0\right] \geq \frac{1}{2}\exp\left(-(1+o(1))\left(n_1 I_r + n_2 I_g + n_3 I_{c1} + n_4 I_{c2}\right)\right). \tag{35}
$$

We refer to Appendix B.1 for the proof of Lemma 5.

**Failure Proof for the Perfect Clustering/Grouping Regime.** Let $\mathcal{T}_\ell$ be a section of columns of $M_0$ with $\tau_\ell = |\mathcal{T}_\ell|/m = \Theta(1)$, and assume $\ell = b_1 b_2 b_3 \in \{0, 1\}^3$. For $c \in \mathcal{T}_\ell$, define $M_{\langle c \rangle}$ be a rating matrix, which is identical to $M_0$, except its $c^{\text{th}}$ column which is given by

$$
\begin{bmatrix} \mathbf{0}_{\frac{n}{6}} & b_1 \mathbf{1}_{\frac{n}{6}} & b_1 \mathbf{1}_{\frac{n}{6}} & b_2 \mathbf{1}_{\frac{n}{6}} & b_3 \mathbf{1}_{\frac{n}{6}} & (b_2 \oplus b_3)\mathbf{1}_{\frac{n}{6}} \end{bmatrix}.
$$

We focus on the family of rating matrices $\{M_{\langle c \rangle} : c \in \mathcal{T}_\ell\}$. It is easy to verify that the type of all such matrices is given by

$$
T = \left(\left\{k_{ij}^{xy} = 0\right\}_{i,j\in[3],\, x,y\in\{A,B\}}, \left\{d_i^A(\ell) = 1\right\}_{i\in\{1,3\},\ell\in\{0,1\}^3}, \left\{d_2^A(\ell) = 0\right\}_{\ell\in\{0,1\}^3},\right.
$$

$$
\left.\left\{d_i^B(\ell) = 0\right\}_{i\in[3],\ell\in\{0,1\}^3}\right). \tag{36}
$$

Using the definition of the negative log-likelihood in (78) for $M_{\langle c \rangle}$ with $c \in \mathcal{T}_\ell$, we obtain

$$
\mathbb{P}\left[\mathsf{L}(M_{\langle c \rangle}) > \mathsf{L}(M_0)\right] = 1 - \mathbb{P}\left[\mathsf{L}(M_{\langle c \rangle}) \leq \mathsf{L}(M_0)\right]
$$

$$
= 1 - \mathbb{P}\left[\log\left(\frac{1-\theta}{\theta}\right)\sum_{i=1}^{\Lambda(M_c, M_0)} B_i^{(p)}\left(2B_i^{(\theta)} - 1\right) \geq 0\right]
$$

$$
= 1 - \mathbb{P}\left[\log\left(\frac{1-\theta}{\theta}\right)\sum_{i=1}^{\frac{n}{3}} B_i^{(p)}\left(2B_i^{(\theta)} - 1\right) \geq 0\right] \tag{37}
$$

$$\leq 1 - \frac{1}{4}\exp\left(-(1+o(1))\frac{n}{3}I_r\right) \tag{38}$$

$$\leq \exp\left(-\frac{1}{4}\exp\left(-(1+o(1))\frac{n}{3}I_r\right)\right), \tag{39}$$

where (37) follows from the evaluation of $\Lambda(M_{\langle c\rangle}, M_0)$ for the type of $M_{\langle c\rangle}$ given in (36), and (38) is an immediate consequence of Lemma 5 by setting $n_1 = \frac{n}{3}$, $n_2 = n_3 = n_4 = 0$.

Next, we can upper bound the success probability of an ML estimator as

$$\mathbb{P}[S] \leq \mathbb{P}\left[\bigcap_{c \in \mathcal{T}_\ell}\left(\mathsf{L}(M_{\langle c\rangle}) > \mathsf{L}(M_0)\right)\right] = \prod_{c \in \mathcal{T}_\ell}\mathbb{P}\left[\mathsf{L}(M_{\langle c\rangle}) > \mathsf{L}(M_0)\right] \tag{40}$$

$$\leq \exp\left(-\frac{1}{4}\exp\left(-(1+o(1))\frac{n}{3}I_r\right)\right)^{\tau_\ell m} \tag{41}$$

$$= \exp\left(-\frac{1}{4}\tau_\ell \exp\left(-(1+o(1))\frac{n}{3}I_r + \log m\right)\right)$$

$$\leq \exp\left(-\frac{1}{4}\tau_\ell \exp\left(-((1+o(1))(1-\epsilon)-1)\log m\right)\right) \tag{42}$$

$$\leq \exp\left(-\frac{1}{4}\tau_\ell \exp\left((\epsilon - o(1)(1-\epsilon))\log m\right)\right), \tag{43}$$

where (40) follows from the fact that the events $\{\mathsf{L}(M_c) > \mathsf{L}(M_0)\}$ are mutually independent for all $c \in \mathcal{T}_\ell$, since each event corresponds to a different column within the block of columns $\mathcal{T}_\ell$; (41) follows from (39); and finally, (42) follows from (25). Therefore, we get

$$\lim_{n,m\to\infty}\mathbb{P}[S] \leq \lim_{n,m\to\infty}\exp\left(-\frac{1}{4}\tau_\ell \exp\left((\epsilon - o(1)(1-\epsilon))\log m\right)\right) = 0, \tag{44}$$

which shows that if (25) holds, then the recovery fails with high probability.

**Failure Proof for the Grouping-Limited Regime.** Without loss of generality, assume $\delta_g m = d_{\mathrm{H}}\left(v_1^A, v_2^A\right)$, i.e., the rating vectors of groups $G_1^A$ and $G_2^A$ that have the minimum inter-group Hamming distance. In the following, we will introduce a class of rating matrices, which are obtained by switching two users between groups $G_1^A$ and $G_2^A$, and prove that if (26) holds, then with high probability the ML estimator will fail by selecting one of the rating matrices from this class, instead of $M_0$.

First, we present the following lemma that guarantees the existence of two subsets of users with certain properties. The proof of the lemma is presented in Appendix B.2.

**Lemma 6.** *Let $\alpha, \beta = \Theta\left(\frac{\log n}{n}\right)$. Consider groups $G_1^A$ and $G_2^A$. As $n \to \infty$, with probability approaching 1, there exists two subgroups $\tilde{G}_1^A \subset G_1^A$ and $\tilde{G}_2^A \subset G_2^A$ with size $|\tilde{G}_1^A| \geq \frac{n}{\log^3 n}$ and $|\tilde{G}_2^A| \geq \frac{n}{\log^3 n}$ such that there is no edge between the nodes in $\tilde{G}_1^A \cup \tilde{G}_2^A$, that is,*

$$E \cap \left((\tilde{G}_1^A \cup \tilde{G}_2^A) \times (\tilde{G}_1^A \cup \tilde{G}_2^A)\right) = \varnothing.$$

For given sub-groups $\tilde{G}_1^A$ and $\tilde{G}_2^A$, we define the set of rating matrices

$$\{M_{\langle f,g\rangle} : f \in \tilde{G}_1^A, g \in \tilde{G}_2^A\}$$

where $M_{\langle f,g\rangle}$ is identical to $M_0$, except its $f^{\text{th}}$ and $g^{\text{th}}$ rows, which are swapped. Note that for every $M_{\langle f,g\rangle}$ in this class, we have $\Lambda(M_{\langle f,g\rangle}, M_0) = 2\delta_g m$. Moreover, the groups induced by $M_{\langle f,g\rangle}$ are $\hat{G}_1^A = G_1^A \cup \{g\} \setminus \{f\}$ and $\hat{G}_2^A = G_2^A \cup \{f\} \setminus \{g\}$, while the other four groups are identical to those of matrix $M_0$. Therefore, for each $M_{\langle f,g\rangle}$ we have

$$\mathsf{L}(M_0) - \mathsf{L}(M_{\langle f,g \rangle})$$

$$= \log\left(\frac{1-\theta}{\theta}\right) \sum_{i=1}^{2\delta_g m} B_i^{(p)}\left(2B_i^{(\theta)} - 1\right)$$

$$+ \log\left(\frac{(1-\beta)\alpha}{(1-\alpha)\beta}\right) \left[ \sum_{h \in G_1^A \setminus \{f\}} \left(B_{(g,h)}^{(\beta)} - B_{(f,h)}^{(\alpha)}\right) + \sum_{h \in G_2^A \setminus \{g\}} \left(B_{(f,h)}^{(\beta)} - B_{(g,h)}^{(\alpha)}\right) \right]$$

$$= \log\left(\frac{1-\theta}{\theta}\right) \sum_{i=1}^{2\delta_g m} B_i^{(p)}\left(2B_i^{(\theta)} - 1\right) + \log\left(\frac{(1-\beta)\alpha}{(1-\alpha)\beta}\right) \sum_{j=1}^{2(\frac{n}{6}-1)} \left(B_j^{(\beta)} - B_j^{(\alpha)}\right)$$

$$= B\left(2\delta_g m, 2(\frac{n}{6} - 1), 0, 0\right)$$

Then, using Lemma 5, we can write

$$\mathbb{P}\left[\mathsf{L}(M_{\langle f,g \rangle}) > \mathsf{L}(M_0)\right] = 1 - \mathbb{P}\left[B(2\delta_g m, 2(\frac{n}{6} - 1), 0, 0) \geq 0\right]$$

$$\leq 1 - \frac{1}{4}\exp\left(-(1+o(1))\left(2\delta_g m I_r + 2\left(\frac{n}{6} - 1\right)I_g\right)\right)$$

$$\leq \exp\left(-\frac{1}{4}\exp\left(-(1+o(1))\left(2\delta_g m I_r + 2\left(\frac{n}{6} - 1\right)I_g\right)\right)\right). \quad (45)$$

Finally, we can bound the success probability of an ML estimator as

$$\mathbb{P}[S] \leq \mathbb{P}\left[\bigcap_{f \in \tilde{G}_1^A, g \in \tilde{G}_2^A} \left(\mathsf{L}(M_{\langle f,g \rangle}) > \mathsf{L}(M_0)\right)\right] = \prod_{f \in \tilde{G}_1^A, g \in \tilde{G}_2^A} \mathbb{P}\left[\mathsf{L}(M_{\langle f,g \rangle}) > \mathsf{L}(M_0)\right] \quad (46)$$

$$\leq \left(\exp\left(-\frac{1}{4}\exp\left(-(1+o(1))\left(2\delta_g m I_r + 2\left(\frac{n}{6} - 1\right)I_g\right)\right)\right)\right)^{|\tilde{G}_1^A| \cdot |\tilde{G}_2^A|} \quad (47)$$

$$= \exp\left(-\frac{n^2}{4\log^6(n)}\exp\left(-(1+o(1))\left(2\delta_g m I_r + 2\left(\frac{n}{6} - 1\right)I_g\right)\right)\right) \quad (48)$$

$$\leq \exp\left(-\frac{n^2}{4\log^6(n)}\exp\left(-2(1+o(1))(1-\epsilon)\log n\right)\right) \quad (49)$$

$$\leq \exp\left(-\frac{n^{2(\epsilon - o(1)(1-\epsilon))}}{4\log^6(n)}\right), \quad (50)$$

where (46) holds since events $\{\mathsf{L}(M_{\langle f,g \rangle}) > \mathsf{L}(M_0)\}$ are independent due to the fact that there is no edge between nodes in $\tilde{G}_1^A \cup \tilde{G}_2^A$; (47) follows from (45); we used $|\tilde{G}_1^A| = |\tilde{G}_2^A| = \frac{n}{\log^3 n}$ in (48); and (49) follows from the condition in (26). Finally, we obtain

$$\lim_{n \to \infty} \mathbb{P}[S] \leq \lim_{n \to \infty} \exp\left(-\frac{n^{2(\epsilon - o(1)(1-\epsilon))}}{4\log^6(n)}\right) = 0,$$

which implies that the ML estimator will fail in finding $M_0$ with high probability.

**Failure Proof for the Clustering-Limited Regime.** The proof of this case follows the same structure as that of the grouping-limited regime. Without loss of generality, assume $v_1^A$ and $v_2^B$ be rating vectors whose minimum hamming distance is $\delta_c m$, i.e., $d_{\mathrm{H}}\left(v_1^A, v_2^B\right) = \delta_c m$. Note that the corresponding groups defined by such rating vectors, $G_1^A$ and $G_2^B$, belong to different clusters. Similar to Lemma 6, we pick subsets $\tilde{G}_1^A \subset G_1^A$ and $\tilde{G}_2^B \subset G_2^B$ with $|\tilde{G}_1^A| = |\tilde{G}_2^B| = \frac{n}{\log^3 n}$. Note that the subgraph induced by $\tilde{G}_1^A \cup \tilde{G}_2^B$ is edge-free. Then, we consider the set of all rating matrices

$$\{M_{\langle f,g \rangle} : f \in \tilde{G}_1^A, g \in \tilde{G}_2^B\},$$

where

$$M_{\langle f,g\rangle}(r,:) = \begin{cases} M_0(g,:) & \text{if } r = f, \\ M_0(f,:) & \text{if } r = g, \\ M_0(r,:) & \text{otherwise.} \end{cases}$$

Then, for $M_{\langle f,g\rangle}$, we have

$$\mathsf{L}(M_0) - \mathsf{L}(M_{\langle f,g\rangle})$$

$$= \log\left(\frac{1-\theta}{\theta}\right)^{\Lambda(M_{\langle f,g\rangle}, M_0)} \sum_{i=1} B_i^{(p)}\left(2B_i^{(\theta)} - 1\right)$$

$$+ \log\left(\frac{(1-\gamma)\alpha}{(1-\alpha)\gamma}\right)\left[\sum_{h\in G_1^A\setminus\{f\}}\left(B_{(g,h)}^{(\gamma)} - B_{(f,h)}^{(\alpha)}\right) + \sum_{h\in G_2^B\setminus\{g\}}\left(B_{(f,h)}^{(\gamma)} - B_{(g,h)}^{(\alpha)}\right)\right]$$

$$+ \log\left(\frac{(1-\gamma)\beta}{(1-\beta)\gamma}\right)\left[\sum_{h\in G_2^A\cup G_3^A}\left(B_{(g,h)}^{(\gamma)} - B_{(f,h)}^{(\beta)}\right) + \sum_{h\in G_1^B\cup G_3^B}\left(B_{(f,h)}^{(\gamma)} - B_{(g,h)}^{(\beta)}\right)\right]$$

$$= B\left(2\delta_c m, 0, 2(\frac{n}{6} - 1), \frac{2n}{3}\right).$$

Applying Lemma 5, we get

$$\mathbb{P}\left[\mathsf{L}(M_{\langle f,g\rangle}) > \mathsf{L}(M_0)\right] = 1 - \mathbb{P}\left[B(2\delta_g m, 2(\frac{n}{6} - 1), 0, 0) \geq 0\right]$$

$$\leq \exp\left(-\frac{1}{4}\exp\left(-(1+o(1))\left(2\delta_c m I_r + 2\left(\frac{n}{6} - 1\right) I_{c1} + 2\frac{n}{3} I_{c2}\right)\right)\right). \tag{51}$$

Therefore, the success probability of the ML estimator can be bounded as

$$\mathbb{P}[S] \leq \prod_{f\in\tilde{G}_1^A, g\in\tilde{G}_2^B} \mathbb{P}\left[\mathsf{L}(M_{\langle f,g\rangle}) > \mathsf{L}(M_0)\right] \tag{52}$$

$$\leq \left(\exp\left(-\frac{1}{4}\exp\left(-(1+o(1))\left(2\delta_c m I_r + 2\left(\frac{n}{6} - 1\right) I_{c1} + 2\frac{n}{3} I_{c2}\right)\right)\right)\right)^{|\tilde{G}_1^A|\cdot|\tilde{G}_2^A|} \tag{53}$$

$$\leq \exp\left(-\frac{n^2}{4\log^6(n)}\exp\left(-2(1+o(1))(1-\epsilon)\log n\right)\right) \tag{54}$$

$$\leq \exp\left(-\frac{n^{2(\epsilon - o(1)(1-\epsilon))}}{4\log^6 n}\right), \tag{55}$$

where (52) is a consequence of independence of the events $\{\mathsf{L}(M_{\langle f,g\rangle}) > \mathsf{L}(M_0)\}$; (53) follows from (51); and in (54) we have used the condition (27). This immediately implies

$$\lim_{n\to\infty}\mathbb{P}[S] = 0,$$

which leads to the failure of the ML estimator.

Since $\lim_{n\to\infty}\mathbb{P}[S] = 0$ is proved under each of the three conditions stated in (25), (26), and (27), the converse proof of Theorem 1 is concluded. $\qquad\square$

# 3 Proof of Theorem 2

**Theorem 2** (Theoretical guarantees of the proposed algorithm). *Assume that $m = \omega(\log n)$, $\log m = o(n)$, $m = O(n)$, $I_{c2} > \frac{2\log n}{n}$ and $I_g > \omega(\frac{1}{n})$. Then, as long as the sample size is beyond the optimal sample complexity in Theorem 1 (i.e., $mnp > mnp^\star$), then the algorithm presented in Section 4 (in the main paper) with $T = O(\log n)$ iterations ensures the worse-case error probability tends to 0 as $n \to \infty$. That is, the algorithm returns $\widehat{M}$ such that $\mathbb{P}[\widehat{M} = M] = 1 - o(1)$.*

*Proof.* We propose a computationally feasible matrix completion algorithm that achieves the optimal sample complexity characterized by Theorem 1. It consists of four phases described as below.

**Phase 1 (Exact Recovery of Clusters):** We use the community detection algorithm in [4] on $\mathcal{G}$ to *exactly*[2] recover the two clusters $A$ and $B$. As proved in [4], the decomposition of the graph into two clusters is correct with high probability when $I_{c2} > \frac{2\log n}{n}$. This completes Phase 1.

**Phase 2 (Almost Exact Recovery of Groups):** The goal of Phase 2 is to decompose the set of users in cluster $A$ (or cluster $B$) into three groups, represented by $G_1^A$, $G_2^A$, $G_3^A$ (or $G_1^B$, $G_2^B$, $G_3^B$). It is worth noting that grouping at this stage is *almost exact*[3], and will be further refined in the next phases. To this end, we run a spectral clustering algorithm [6] on $A$ and $B$ separately. Let $\widehat{G}_i^x(0)$ denote the initial estimate of the $i^{\text{th}}$ group of cluster $x$ that is recovered by Phase 2, for $i \in [3]$ and $x \in \{A, B\}$. It is shown that the groups within each cluster are recovered with a vanishing fraction of errors if $I_g = \omega(1/n)$. It is worth mentioning that there are other clustering algorithms [7–14] that can be employed for this phase. Examples include: spectral clustering [7–11], semidefinite programming (SDP) [12], non-backtracking matrix spectrum [13], and belief propagation [14]. This completes Phase 2.

**Phase 3 (Exact Recovery of Rating Vectors):** We propose a novel algorithm that optimally recovers the rating vectors of the groups within each cluster. The algorithm is based on maximum likelihood (ML) decoding of users' ratings based on the partial and noisy observations. For this model, the ML decoding boils down to a counting rule: for each item, find the group with the maximum gap between the number of observed zeros and ones, and set the rating entry of this group to 0. The other two rating vectors are either both 0 or both 1 for this item, which will be determined based on the majority of the union of their observed entries. It turns out that the vector recovery is exact with probability $1 - o(1)$. We first present the proposed algorithm. Then, the theoretical guarantee of the algorithm is provided.

Define $\widehat{v}_i^x$ as the estimated rating vector of $v_i^x$, i.e., the output of Algorithm 1 (see below). Let the $c^{\text{th}}$ element of the rating vector $v_i^x$ (or $\widehat{v}_i^x$) be denoted by $v_i^x(c)$ (or $\widehat{v}_i^x(c)$) for $i \in [3]$, $x \in \{A, B\}$ and $c \in [m]$. Let $Y_{r,c}$ be the $(r, c)$-entry of matrix $Y$, and $Z_{r,c}$ be its mapping to $\{+1, 0, -1\}$ for $r \in [n]$ and $c \in [m]$. The pseudocode is given below.

---
**Algorithm 1** Exact Recovery of Rating Vectors

---
1: **function** VECRCV $(n, m, Z, \{\widehat{G}_i^x(0) : i \in [3], x \in \{A, B\}\})$
2:     **for** $c \in [m]$ and $x \in \{A, B\}$ **do**
3:         **for** $i \in [3]$ **do** $\rho_{i,x}(c) \leftarrow \sum_{r \in \widehat{G}_i^x(0)} Z_{r,c}$
4:         $j \leftarrow \arg\max_{i \in [3]} \rho_{i,x}(c)$
5:         $\widehat{v}_j^x(c) \leftarrow 0$
6:         **if** $\sum_{i \in [3] \setminus \{j\}} \rho_{i,x}(c) \geq 0$ **then**
7:             **for** $i \in [3] \setminus \{j\}$ **do** $\widehat{v}_i^x(c) \leftarrow 0$
8:         **else**
9:             **for** $i \in [3] \setminus \{j\}$ **do** $\widehat{v}_i^x(c) \leftarrow 1$
10:    **return** $\{\widehat{v}_i^x : i \in [3], x \in \{A, B\}\}$

**Remark 2.** *Algorithm 1 is one of the technical distinctions, relative to the prior works [2, 3] which employ the simple majority voting rule under non-hierarchical SBMs. Also our technical novelty in analysis, reflected in (57) (see below), exploits the hierarchical structure to prove the theoretical guarantee.* ∎

Let us now prove the exact recovery of the rating vectors of the groups within cluster $A$. The proof w.r.t. cluster $B$ follows by symmetry. Without loss of generality, assume that $v_1^A(c) = 0$ for $c \in [m/2]$, and $v_1^A(c) = 1$ for $c \in m \setminus [m/2]$. In what follows, we will prove that $v_1^A$ can be exactly recovered, i.e., $\mathbb{P}\left[\widehat{v}_1^A = v_1^A\right] = 1 - o(1)$. Similar proofs can be constructed for $v_2^A$ and $v_3^A$. The probability of error in recovering $v_1^A$ is expressed as

$$\mathbb{P}\left[\widehat{v}_1^A \neq v_1^A\right]$$

$$= \mathbb{P}\left[\left(\bigcup_{c \in [m/2]} \{\widehat{v}_1^A(c) = 1\}\right) \cup \left(\bigcup_{c \in m \setminus [m/2]} \{\widehat{v}_1^A(c) = 0\}\right)\right]$$

$$\leq \left(\sum_{c \in [m/2]} \mathbb{P}\left[\widehat{v}_1^A(c) = 1\right]\right) + \left(\sum_{c \in m \setminus [m/2]} \mathbb{P}\left[\widehat{v}_1^A(c) = 0\right]\right) \tag{56}$$

$$= \left(\sum_{c \in [m/2]} \mathbb{P}\left[\bigcap_{i \in [2]} \{\widehat{v}_i^A(c) = 1\} \cap \{\widehat{v}_3^A(c) = 0\}\right] + \mathbb{P}\left[\bigcap_{i \in \{1,3\}} \{\widehat{v}_i^A(c) = 1\} \cap \{\widehat{v}_2^A(c) = 0\}\right]\right)$$

$$+ \left(\sum_{c \in m \setminus [m/2]} \mathbb{P}\left[\bigcap_{i \in [3]} \{\widehat{v}_i^A(c) = 0\}\right] + \mathbb{P}\left[\{\widehat{v}_1^A(c) = 0\} \cap \bigcap_{i \in \{2,3\}} \{\widehat{v}_i^A(c) = 1\}\right]\right) \tag{57}$$

$$\leq \left(\sum_{c \in [m/2]} \mathbb{P}\left[\rho_{1,A}(c) + \rho_{2,A}(c) \leq 0\right] + \sum_{c \in [m/2]} \mathbb{P}\left[\rho_{1,A}(c) + \rho_{3,A}(c) \leq 0\right]\right)$$

$$+ \left(\sum_{c \in m \setminus [m/2]} \mathbb{P}\left[\rho_{2,A}(c) + \rho_{3,A}(c) \geq 0\right] + \sum_{c \in m \setminus [m/2]} \mathbb{P}\left[\rho_{2,A}(c) + \rho_{3,A}(c) \geq 0\right]\right) \tag{58}$$

$$= \left(\sum_{c \in [m/2]} \mathbb{P}\underbrace{\left[\sum_{r_1 \in \widehat{G}_1^A(0)} Z_{r_1 c} + \sum_{r_2 \in \widehat{G}_2^A(0)} Z_{r_2 c} \leq 0\right]}_{\text{Term}_1} + \sum_{c \in [m/2]} \mathbb{P}\underbrace{\left[\sum_{r_1 \in \widehat{G}_1^A(0)} Z_{r_1 c} + \sum_{r_3 \in \widehat{G}_3^A(0)} Z_{r_3 c} \leq 0\right]}_{\text{Term}_2}\right)$$

$$+ \left(\sum_{c \in m \setminus [m/2]} \mathbb{P}\underbrace{\left[\sum_{r_2 \in \widehat{G}_2^A(0)} Z_{r_2 c} + \sum_{r_3 \in \widehat{G}_3^A(0)} Z_{r_3 c} \geq 0\right]}_{\text{Term}_3} + \sum_{c \in m \setminus [m/2]} \mathbb{P}\underbrace{\left[\sum_{r_2 \in \widehat{G}_2^A(0)} Z_{r_2 c} + \sum_{r_3 \in \widehat{G}_3^A(0)} Z_{r_3 c} \geq 0\right]}_{\text{Term}_4}\right) \tag{59}$$

where (56) follows from the union bound; (57) follows from $v_1^A \oplus v_2^A = v_3^A$; (58) follows from the ML decoding outlined in Algorithm 1; and (59) follows from the definition of $\rho_{i,x}(c)$ on Line 3 in Algorithm 1.

Next we show that each of the four terms in (59) is $o(m^{-1})$. We prove that for $\text{Term}_1$ and $\text{Term}_3$, and similar proofs can be carried out for $\text{Term}_2$ and $\text{Term}_4$. Define $R_i := \widehat{G}_i^A(0) \setminus G_i^A$ and $\eta_i := |R_i| / n$. From the theoretical guarantees (i.e., exact clustering and almost-exact grouping) in Phases 1 and 2, we have $\lim_{n \to \infty} \eta_i = 0$, $\forall i \in [3]$ with high probability. Define $n_{i1} := \left(\frac{1}{6} - \eta_i\right) n$ and $n_{i2} := \eta_i n$ for $i \in [3]$. Let $\{B_i^{(p)}\} \overset{\text{i.i.d.}}{\sim} \text{Bern}(p)$, and $\{B_i^{(\theta)}\} \overset{\text{i.i.d.}}{\sim} \text{Bern}(\theta)$. Hence, for $c \in [m/2]$, $\text{Term}_1$ can be

upper bounded by

$$\mathbb{P}\left[\sum_{r_1\in\widehat{G}_1^A(0)}Z_{r_1c}+\sum_{r_2\in\widehat{G}_2^A(0)}Z_{r_2c}\leq 0\right]$$

$$=\mathbb{P}\left[\sum_{i\in\widehat{G}_1^A(0)\setminus R_1}Z_{ic}+\sum_{j\in R_1}Z_{jc}+\sum_{k\in\widehat{G}_2^A(0)\setminus R_2}Z_{kc}+\sum_{\ell\in R_2}Z_{\ell c}\leq 0\right]$$

$$\leq\mathbb{P}\left[\sum_{i\in\widehat{G}_1^A(0)\setminus R_1}Z_{ic}-\sum_{j\in R_1}|Z_{jc}|+\sum_{k\in\widehat{G}_2^A(0)\setminus R_2}Z_{kc}-\sum_{\ell\in R_2}|Z_{\ell c}|\leq 0\right]$$

$$=\mathbb{P}\left[-\sum_{i=1}^{n_{11}}B_i^{(p)}\left(2B_i^{(\theta)}-1\right)-\sum_{j=n_{11}+1}^{n_{11}+n_{12}}B_j^{(p)}\right.$$

$$\left.-\sum_{k=n_{11}+n_{12}+1}^{n_{11}+n_{12}+n_{21}}B_k^{(p)}\left(2B_k^{(\theta)}-1\right)-\sum_{\ell=n_{11}+n_{12}+n_{21}+1}^{n_{11}+n_{12}+n_{21}+n_{22}}B_\ell^{(p)}\leq 0\right] \tag{60}$$

$$=\mathbb{P}\left[\sum_{i=1}^{n_{11}+n_{21}}B_i^{(p)}\left(2B_i^{(\theta)}-1\right)\geq -\sum_{j=n_{11}+n_{21}+1}^{n_{11}+n_{21}+n_{12}+n_{22}}B_j^{(p)}\right] \tag{61}$$

where (60) follows since $v_1^A(c)=0$ for $c\in[m/2]$,

$$Y_{jc}=\begin{cases} 0 & \text{w.p. } p(1-\theta); \\ 1 & \text{w.p. } p\theta, \end{cases}$$

and $Z_{jc}=-(2Y_{jc}-1)$.

The following lemma introduces a large deviation result employed in [2] to further bound (61).

**Lemma 7.** *Let $0<\epsilon<1$, and $0<p<1/2$. Suppose $X\sim\mathsf{Binom}(\epsilon n,p)$. Then,*

$$\mathbb{P}\left[X\geq\frac{\kappa np}{\log(1/\epsilon)}\right]\leq 2\exp\left(-\frac{\kappa np}{2}\right), \quad \text{for any } \kappa\geq 2e. \tag{62}$$

*Proof.* The proof is given by [2, Lemma 7]. □

Let $\kappa$ be sufficiently large such that $\kappa>4e$. Thus, the RHS of (61) can be upper bounded by

$$\mathbb{P}\left[\sum_{r_1\in\widehat{G}_1^A(0)}Z_{r_1c}+\sum_{r_2\in\widehat{G}_2^A(0)}Z_{r_2c}\leq 0\right]$$

$$\leq\mathbb{P}\left[\sum_{i=1}^{n_{11}+n_{21}}B_i^{(p)}\left(2B_i^{(\theta)}-1\right)\geq -\sum_{j=n_{11}+n_{21}+1}^{n_{11}+n_{21}+n_{12}+n_{22}}B_j^{(p)}\right]$$

$$\leq\mathbb{P}\left[\sum_{i=1}^{n_{11}+n_{21}}B_i^{(p)}\left(2B_i^{(\theta)}-1\right)\geq -\frac{\kappa np}{\log\frac{1}{\eta_1+\eta_2}}\right]+\mathbb{P}\left[-\sum_{j=n_{11}+n_{21}+1}^{n_{11}+n_{21}+n_{12}+n_{22}}B_j^{(p)}\leq -\frac{\kappa np}{\log\frac{1}{\eta_1+\eta_2}}\right]$$

$$\leq\mathbb{P}\left[\sum_{i=1}^{n_{11}+n_{21}}B_i^{(p)}\left(2B_i^{(\theta)}-1\right)\geq -\frac{\kappa np}{\log\frac{1}{\eta_1+\eta_2}}\right]+2\exp\left(-\frac{\kappa np}{2}\right) \tag{63}$$

$$\leq\mathbb{P}\left[\log\left(\frac{1-\theta}{\theta}\right)\sum_{i=1}^{n_{11}+n_{21}}B_i^{(p)}\left(2B_i^{(\theta)}-1\right)\geq -\log\left(\frac{1-\theta}{\theta}\right)\frac{cnp}{\log\frac{1}{\eta_1+\eta_2}}\right]+o(m^{-1}) \tag{64}$$

$$\leq\exp\left(\frac{1}{2}\log\left(\frac{1-\theta}{\theta}\right)\frac{cnp}{\log\frac{1}{\eta_1+\eta_2}}-(1+o(1))\left(\frac{1}{3}-(\eta_1+\eta_2)\right)nI_r\right)+o(m^{-1}) \tag{65}$$

$$\approx \exp\left(-(1+o(1))\left(\frac{1}{3} - (\eta_1 + \eta_2)\right) n I_r\right) + o(m^{-1}) \tag{66}$$

$$\leq \exp\left(-(1+o(1))\left(1 + \frac{\epsilon}{4}\right) \log m\right) + o(m^{-1}) \tag{67}$$

$$= o(m^{-1}) \tag{68}$$

where (63) follows from Lemma 7; (64) follows since $np = \Omega(\log m)$; (65) readily follows from Lemma 2; (66) follows as the first term in the exponent is insignificant compared to the other term since $np = \Theta(nI_r)$ and $\lim_{\eta_1,\eta_2 \to 0^+} \frac{1}{\log \frac{1}{\eta_1+\eta_2}} = 0$; and (67) follows since $\frac{1}{3}nI_r \geq (1 + \epsilon) \log m$ guarantees that $\left(\frac{1}{3} - (\eta_1 + \eta_2)\right) nI_r \geq \left(1 + \frac{\epsilon}{4}\right) \log m$ as long as $(\eta_1 + \eta_2)$ is sufficiently small compared to $\epsilon$.

Similarly, for $c \in m \setminus [m/2]$, $\mathrm{Term}_3$ can be upper bounded by

$$\mathbb{P}\left[\sum_{r_2 \in \widehat{G}_2^A(0)} Z_{r_2 c} + \sum_{r_3 \in \widehat{G}_3^A(0)} Z_{r_3 c} \geq 0\right]$$

$$= \mathbb{P}\left[\sum_{i \in \widehat{G}_2^A(0) \setminus R_2} Z_{ic} + \sum_{j \in R_2} Z_{jc} + \sum_{k \in \widehat{G}_3^A(0) \setminus R_3} Z_{kc} + \sum_{\ell \in R_3} Z_{\ell c} \geq 0\right]$$

$$\leq \mathbb{P}\left[\sum_{i \in \widehat{G}_2^A(0) \setminus R_2} Z_{ic} + \sum_{j \in R_2} |Z_{jc}| + \sum_{k \in \widehat{G}_3^A(0) \setminus R_3} Z_{kc} + \sum_{\ell \in R_3} |Z_{\ell c}| \geq 0\right]$$

$$= \mathbb{P}\left[\sum_{i=1}^{n_{21}} B_i^{(p)}\left(2B_i^{(\theta)} - 1\right) + \sum_{j=n_{21}+1}^{n_{21}+n_{22}} B_j^{(p)}\right.$$

$$\left. + \sum_{k=n_{21}+n_{22}+1}^{n_{21}+n_{22}+n_{31}} B_k^{(p)}\left(2B_k^{(\theta)} - 1\right) + \sum_{\ell=n_{21}+n_{22}+n_{31}+1}^{n_{21}+n_{22}+n_{31}+n_{32}} B_\ell^{(p)} \geq 0\right] \tag{69}$$

$$= \mathbb{P}\left[\sum_{i=1}^{n_{21}+n_{31}} B_i^{(p)}\left(2B_i^{(\theta)} - 1\right) \geq - \sum_{j=n_{21}+n_{31}+1}^{n_{21}+n_{31}+n_{22}+n_{32}} B_j^{(p)}\right] \tag{70}$$

where (69) follows since $v_1^A(c) = 1$ for $c \in m \setminus [m/2]$,

$$Y_{jc} = \begin{cases} 0 & \text{w.p. } p\theta; \\ 1 & \text{w.p. } p(1-\theta), \end{cases}$$

and $Z_{jc} = -(2Y_{jc} - 1)$. Applying similar bounding techniques used for (60), one can show that

$$\mathbb{P}\left[\sum_{r_2 \in \widehat{G}_2^A(0)} Z_{r_2 c} + \sum_{r_3 \in \widehat{G}_3^A(0)} Z_{r_3 c} \geq 0\right] \leq o(m^{-1}). \tag{71}$$

Finally, by (68) and (71), the probability of error in recovering $v_1^A$ is upper bounded by

$$\mathbb{P}\left[\widehat{v}_1^A \neq v_1^A\right]$$

$$\leq \left(\sum_{c \in [m/2]} o(m^{-1}) + \sum_{c \in [m/2]} o(m^{-1})\right) + \left(\sum_{c \in m \setminus [m/2]} o(m^{-1}) + \sum_{c \in m \setminus [m/2]} o(m^{-1})\right)$$

$$= o(1). \tag{72}$$

This completes the proof of exact recovery of rating vectors.

**Phase 4 (Exact Recovery of Groups):** The goal in this last step is to *refine* the groups which are *almost recovered* in Phase 2, thereby obtaining an *exact* grouping. To this end, we propose an

iterative algorithm that locally refines the estimates on the user grouping within each cluster for $T$ iterations. More specifically, at each iteration, the affiliation of each user is updated to the group that yields the maximum point-wise likelihood w.r.t. the considered user. The exact computation of the point-wise likelihood requires the knowledge of the model parameters $(\alpha, \beta, \theta)$. But we do not rely on such knowledge, instead estimate them using the given ratings and graph $(Y, \mathcal{G})$. Hence, we use an *approximated* point-wise log-likelihood which can readily be computed as:

$$|\{c: Y_{r,c} = \widehat{v}_i^x(c)\}| \cdot \log\left(\frac{1-\widehat{\theta}}{\widehat{\theta}}\right) + e\left(\{r\}, \widehat{G}_i^x(t-1)\right) \cdot \log\left(\frac{(1-\widehat{\beta})\widehat{\alpha}}{(1-\widehat{\alpha})\widehat{\beta}}\right) \tag{73}$$

where $(\widehat{\alpha}, \widehat{\beta}, \widehat{\theta})$ denote the maximum likelihood estimates of $(\alpha, \beta, \theta)$. Here $|\{c: Y_{r,c} = \widehat{v}_i^x(c)\}|$ indicates the number of observed rating matrix entries of the user that coincide with the corresponding entries of the rating vector of that group; and $e\left(\{r\}, \widehat{G}_i^x(t-1)\right)$ denotes the number of edges between the user and the set of users which belong to that group. The pseudocode is described in Algorithm 2.

---

**Algorithm 2** Local Iterative Refinement of Groups (Set $flag = 0$)

1: **function** REFINE $(flag, n, m, T, Y, \mathcal{G}, \{(\widehat{G}_i^x(0), \widehat{v}_i^x) : i \in [3], x \in \{A, B\}\})$
2:      $\widehat{\alpha} \leftarrow \frac{1}{6\binom{n/6}{2}} |\{(f,g) \in E : f, g \in G_i^x, x \in \{A, B\}, i \in [3]\}|$
3:      $\widehat{\beta} \leftarrow \frac{6}{n^2} |\{(f,g) \in E : f \in G_i^x, g \in G_j^x, x \in \{A, B\}, i \in [3], j \in [3] \setminus i\}|$
4:      $\widehat{\theta} \leftarrow |\{(r,c) \in \Omega : Y_{rc} \neq \widehat{v}_i^x(c), r \in \widehat{G}_i^x(0)\}|/|\Omega|$
5:      **for** $t \in [T]$ and $x \in \{A, B\}$ **do**
6:          **for** $i \in [3]$ **do** $\widehat{G}_i^x(t) \leftarrow \varnothing$
7:          **for** $r \leftarrow 1$ **to** $n$ **do**
8:              $j \leftarrow \arg\max_{i \in [3]} |\{c: Y_{r,c} = \widehat{v}_i^x(c)\}| \cdot \log\left(\frac{1-\widehat{\theta}}{\widehat{\theta}}\right) + e\left(\{r\}, \widehat{G}_i^x(t-1)\right) \cdot \log\left(\frac{(1-\widehat{\beta})\widehat{\alpha}}{(1-\widehat{\alpha})\widehat{\beta}}\right)$
9:              $\widehat{G}_j^x(t) \leftarrow \widehat{G}_j^x(t) \cup \{r\}$
10:              **if** $flag == 1$ **then**
11:                  $\{\widehat{v}_i^x : i \in [3], x \in \{A, B\}\} \leftarrow$ VECRCV $(n, m, Y, \{\widehat{G}_i^x(t) : i \in [3], x \in \{A, B\}\})$
12:      **return** $\{\widehat{G}_i^x(T) : i \in [3], x \in \{A, B\}\}, \{\widehat{v}_i^x : i \in [3], x \in \{A, B\}\}$

---

In order to prove that Algorithm 2 ensures the exact recovery of groups, we intend to show that the number of misclassified users in each cluster strictly decreases with each iteration. To this end, we rely on a technique that was employed in many relevant papers [2, 3, 15]. The technique aims to prove that the misclassification error rate is reduced by a factor of 2 with each iteration. More specifically, assuming that the previous phases are executed successfully, if we start with $\eta n$ misclassified users within one cluster, for some small $\eta > 0$, then it intends to show that we end up with $\frac{\eta}{2} n$ misclassified users with high probability as $n \to \infty$ after one iteration of refinement. Hence, with this technique, running the local refinement for $T = \frac{\log(\eta n)}{\log 2}$ within the groups of each cluster would suffice to converge to the ground truth assignments. The proof of such error rate reduction follows the one in [3, Theorem 2] in which the problem of recovering $K$ communities of possibly different sizes is studied. By considering the case of three equal-sized communities, the guarantees of exact recovery of the groups within each cluster readily follows when $T = O(\log n)$.

**Remark 3.** *The iterative refinement in Algorithm 2 can be applied only on the groups (when $flag = 0$), or on the groups as well as the rating vectors (for $flag = 1$). Even though the former is sufficient for reliable estimation of the rating matrix, we show, through our simulation results in the following section, that the latter achieves a better performance for finite regimes of $n$ and $m$.* ∎

This completes the proof of Phase 4, and concludes the proof of Theorem 2. □

# 4 Supplementary Experimental Results

Similar to [2, 3, 16, 17], the performance of the proposed algorithm is assessed on semi-real data (real graph but synthetic rating vectors). We consider a subgraph of the political blog network [18], which is shown to exhibit a hierarchical structure [19]. In particular, we consider a tall matrix setting of $n = 381$ and $m = 200$ in order to investigate the gain in sample complexity due to the graph side information. The selected subgraph consists of two clusters of political parties, each of which comprises three groups. The three groups of the first cluster consist of 98, 34 and 103 users, while the three groups of the second cluster consist of 58, 68 and 20 users.

In order to visualize the underlying hierarchical structure of the considered subgraph of the political blog network, we apply a dimensionality reduction algorithm, called t-Distributed Stochastic Neighbor Embedding (t-SNE) [20] to visualize high-dimensional data in a low-dimensional space. Fig. 1 shows two clusters that are colored in red and blue. Each cluster comprises three groups, represented by circle, triangle and square.

Figure 1: Visualization of a subgraph of the political blog network [18] using t-SNE algorithm [20].

# A Proofs of Lemmas for Achievability Proof of Theorem 1

## A.1 Proof of Lemma 1

We prove that the set $\mathcal{T}^{(\delta)}$ of tuples, characterized by (13), is sufficient to fully represent all $X \in \mathcal{M}^{(\delta)}$. It should be noted that, for a fixed $G_i^x$, one can interpret $\sum_{y \in \{A,B\}} \sum_{j \in [3]} k_{ij}^{xy}$ as the number of users in $G_i^x$ whose rating vectors are swapped from $v_i^x$ to other rating vectors.

Suppose there exists a group $G_i^x$ such that $\sum_{y \in \{A,B\}} \sum_{j \in [3]} k_{ij}^{xy} > \frac{5n}{36}$. This implies that the number of users in $G_i^x$ whose rating vectors are unaltered is less than or equal to $\frac{n}{36}$. Note that the size of each group is $n/6$, and the group sizes must be conserved that for any $X \in \mathcal{M}^{(\delta)}$. Consequently, there must be users whose rating vectors are swapped from other rating vectors to $v_i^x$, and the number of such users is given by

$$\sum_{y \in [c]} \sum_{j \in [g]} k_{ji}^{yx} = \sum_{y \in [c]} \sum_{j \in [g]} k_{ij}^{xy} > \frac{5n}{36}. \tag{74}$$

Since there are 5 groups other than $G_i^x$, hence, by (74), there exists at least one group $G_j^y$ such that

$$k_{ji}^{yx} \geq \frac{n}{36}, \tag{75}$$

where the LHS of (75) gives the number of users in such a group $G_j^y$ whose rating vectors are swapped from $v_j^y$ of such $G_j^y$ to $v_i^x$. Switch the roles of $G_i^x$ and $G_j^y$. Hence, the number of users in $G_i^x$ whose rating vectors are unaltered is larger than $\frac{n}{36}$, which implies that

$$\sum_{y \in [c]} \sum_{j \in [g]} k_{ij}^{xy} \leq \frac{5n}{36}, $$

as per (13). This completes the proof of Lemma 1. $\qquad\square$

## A.2 Proof of Lemma 2

We will first calculate $\mathsf{L}(X)$. Let $e_g(X)$ be the number of edges between groups within clusters and $e_c(X)$ be the number of edges across clusters w.r.t. a rating matrix $X$. Then, we get

$$\mathbb{P}[Y \mid X] = (1-p)^{|\Omega|} p^{nm-|\Omega|}(1-\theta)^{|\Omega|-\Lambda(Y,X)}\theta^{\Lambda(Y,X)}, \tag{76}$$

$$\mathbb{P}[\mathcal{G} \mid X] = \gamma^{e_c(X)}(1-\gamma)^{(\frac{n}{2})^2-e_c(X)}\beta^{e_g(X)}(1-\beta)^{6(\frac{n}{6})^2-e_g(X)}$$
$$\cdot \alpha^{|E|-e_g(X)-e_c(X)}(1-\alpha)^{6\binom{n/6}{2}-(|E|-e_g(X)-e_c(X))} \tag{77}$$

where $|\Omega|$ indicates the number of observed entries and $\Lambda(Y,X)$ denotes the number of distinct entries between $Y$ and $X$. By (76) and (77),

$$\mathsf{L}(X) = -\log \mathbb{P}[Y \mid X] - \log \mathbb{P}[\mathcal{G} \mid X]$$
$$= \log\left(\frac{1-\theta}{\theta}\right)\Lambda(Y,X) + \log\left(\frac{(1-\beta)\alpha}{(1-\alpha)\beta}\right)e_g(X) + \log\left(\frac{(1-\gamma)\alpha}{(1-\alpha)\gamma}\right)e_c(X) + c$$
$$= c_1\Lambda(Y,X) + c_2 e_g(X) + c_3 e_c(X) + c \tag{78}$$

where $c_1 := \log\left(\frac{1-\theta}{\theta}\right)$, $c_2 := \log\left(\frac{(1-\beta)\alpha}{(1-\alpha)\beta}\right)$, $c_3 := \log\left(\frac{(1-\gamma)\alpha}{(1-\alpha)\gamma}\right)$, and $c$ is some constant which is irrelevant to $X$.

By (78), $\mathsf{L}(X) - \mathsf{L}(M_0)$ can be written as

$$c_1(\Lambda(Y,X) - \Lambda(Y,M_0)) + c_2(e_g(X) - e_g(M_0)) + c_3(e_c(X) - e_c(M_0)). \tag{79}$$

Since $\Lambda(Y,X)$ indicates the number of distinct entries between $Y$ and $X$, its mathematical definition reads

$$\Lambda(Y,X) := |\{(r,c) : (Y)_{rc} \neq (X)_{rc}\}| = \sum_{(r,c) \in \Omega} \mathbb{1}\{(Y)_{rc} \neq (X)_{rc}\}.$$

Thus, $\Lambda(Y,X) - \Lambda(Y,M_0)$ can be computed as

$$
\begin{aligned}
\Lambda(Y,X) - \Lambda(Y,M_0) &= \sum_{(r,c)\in\Omega} \mathbb{1}\left\{(Y)_{rc} \neq (X)_{rc}\right\} - \sum_{(r,c)\in\Omega} \mathbb{1}\left\{(Y)_{rc} \neq (M_0)_{rc}\right\} \\
&= \sum_{\substack{(r,c)\in\Omega: \\ (M_0)_{rc}\neq(X)_{rc}}} \left[\mathbb{1}\left\{(Y)_{rc} = (M_0)_{rc}\right\} - \mathbb{1}\left\{(Y)_{rc} = (X)_{rc}\right\}\right] \\
&= \sum_{l=1}^{\Lambda(M_0,X)} \left[B_l^{(p)}\left(1 - B_l^{(\theta)}\right) - B_l^{(p)}B_l^{(\theta)}\right] \\
&= \sum_{l=1}^{\Lambda(M_0,X)} \left[B_l^{(p)}\left(1 - 2B_l^{(\theta)}\right)\right].
\end{aligned}
\tag{80}
$$

Furthermore, $\Lambda(M_0, X) = |\{(r,c) : (M_0)_{rc} \neq (X)_{rc}\}|$ reads

$$
\begin{aligned}
\Lambda(M_0, X) := &\left[\sum_{x\in\{A,B\}} \sum_{i\in[3]} \sum_{y\in\{A,B\}} \sum_{j\in[3]} k_{ij}^{xy} \, d_{\mathrm{H}}\left(v_i^x, u_j^y\right)\right] \\
&+ \left[\sum_{x\in\{A,B\}} \sum_{i\in[3]} \left(\frac{n}{6} - \sum_{y\in\{A,B\}} \sum_{j\in[3]} k_{ij}^{x[3]}\right)\left(\sum_{\ell\in\{0,1\}^3} d_i^x(\ell)\right)\right],
\end{aligned}
\tag{81}
$$

where $d_{\mathrm{H}}\left(v_i^x, u_j^y\right)$ denotes the hamming distance between two vectors $v_i^x$ and $u_j^y$.

We decompose vectors into $\ell$-blocks. The vector $v_i^x(\ell)$ is an either all-one or all-zero vector, for every choice of $(x,i,\ell)$. Hence, $d_{\mathrm{H}}\left(v_i^x(\ell), v_j^y(\ell)\right)$ is either $0$ or $\delta_\ell$. Therefore, $d_{\mathrm{H}}\left(v_i^x, u_j^y\right)$ can be written as

$$
\begin{aligned}
d_{\mathrm{H}}\left(v_i^x, u_j^y\right) &= \sum_{\ell\in\{0,1\}^3} d_{\mathrm{H}}\left(v_i^x(\ell), u_j^y(\ell)\right) \\
&= \sum_{\ell\in\Delta(v_i^x,v_j^y)} d_{\mathrm{H}}\left(v_i^x(\ell), u_j^y(\ell)\right) + \sum_{\ell\notin\Delta(v_i^x,v_j^y)} d_{\mathrm{H}}\left(v_i^x(\ell), u_j^y(\ell)\right) \\
&\overset{(a)}{=} \sum_{\ell\in\Delta(v_i^x,v_j^y)} d_{\mathrm{H}}\left(\mathbf{1}_{1\times\delta_\ell} \oplus v_j^y(\ell), u_j^y(\ell)\right) + \sum_{\ell\notin\Delta(v_i^x,v_j^y)} d_{\mathrm{H}}\left(v_j^y(\ell), u_j^y(\ell)\right) \\
&= \sum_{\ell\in\Delta(v_i^x,v_j^y)} \left(\delta_\ell - d_{\mathrm{H}}\left(v_j^y(\ell), u_j^y(\ell)\right)\right) + \sum_{\ell\notin\Delta(v_i^x,v_j^y)} d_{\mathrm{H}}\left(v_j^y(\ell), u_j^y(\ell)\right) \\
&= \sum_{\ell\in\Delta(v_i^x,v_j^y)} \delta_\ell - \sum_{\ell\in\Delta(v_i^x,v_j^y)} d_j^y(\ell) + \sum_{\ell\notin\Delta(v_j^y,v_j^y)} d_j^y(\ell) \\
&= \sum_{\ell\in\{0,1\}^3} d_{\mathrm{H}}\left(v_i^x(\ell), v_j^y(\ell)\right) - \sum_{\ell\in\Delta(v_i^x,v_j^y)} d_j^y(\ell) + \sum_{\ell\notin\Delta(v_j^y,v_j^y)} d_j^y(\ell) \\
&= d_{\mathrm{H}}\left(v_i^x, v_j^y\right) - \sum_{\ell\in\Delta(v_i^x,v_j^y)} d_j^y(\ell) + \sum_{\ell\notin\Delta(v_j^y,v_j^y)} d_j^y(\ell),
\end{aligned}
\tag{82}
$$

where $\Delta(v_i^x, v_j^y)$ indicates the set of subscripts of indices of the column blocks at which the rating vectors $v_i^x$ and $v_j^y$ differ where

$$
\Delta(v_i^x, v_j^y) = \{\ell \in \{0,1\}^3 : v_i^x(\ell) \neq v_j^y(\ell)\}.
\tag{83}
$$

Note that $(a)$ holds since whenever $\ell \notin \Delta(v_i^x, v_j^y)$ we have $v_i^x(\ell) = v_j^y(\ell)$, and $\ell \in \Delta(v_i^x, v_j^y)$ implies $v_i^x(\ell)$ and $v_j^y(\ell)$ are different in all positions. Thus, (81) can be written as

$$\Lambda(M_0, X)$$

$$:= \left[ \sum_{x \in \{A,B\}} \sum_{i \in [3]} \sum_{y \in \{A,B\}} \sum_{j \in [3]} k_{ij}^{xy} \left( d_{\mathrm{H}} \left( v_i^x, v_j^y \right) - \sum_{\ell \in \Delta(v_i^x, v_j^y)} d_j^x(\ell) + \sum_{\ell \in \{0,1\}^3 \setminus \Delta(v_i^x, v_j^y)} d_j^x(\ell) \right) \right]$$

$$+ \left[ \sum_{x \in \{A,B\}} \sum_{i \in [3]} \left( \frac{n}{6} - \sum_{y \in \{A,B\}} \sum_{j \in [3]} k_{ij}^{xy} \right) \left( \sum_{\ell \in \{0,1\}^3} d_i^x(\ell) \right) \right]. \tag{84}$$

We will now evaluate $e_g(X) - e_g(M_0)$ and $e_c(X) - e_c(M_0)$. Let us denote $x$-type edges that appear with probability $x$ between users where $x \in \{\alpha, \beta, \gamma\}$. Then, $e_g(X) - e_g(M_0)$ denotes the difference between the number of $\beta$-type edges for $X$ and that of $M_0$, while $e_c(X) - e_c(M_0)$ denotes the difference between the number of $\gamma$-type edges for $X$ and that of $M_0$ where

$$e_g(X) - e_g(M_0) = \sum_{i=1}^{\eta_T^{\alpha \to \beta}} B_i^{(\alpha)} + \sum_{i=1}^{\eta_T^{\gamma \to \beta}} B_i^{(\gamma)} - \sum_{i=1}^{\eta_T^{\beta \to \alpha} + \eta_T^{\beta \to \gamma}} B_i^{(\beta)}, \tag{85}$$

$$e_c(X) - e_c(M_0) = \sum_{i=1}^{\eta_T^{\alpha \to \gamma}} B_i^{(\alpha)} + \sum_{i=1}^{\eta_T^{\beta \to \gamma}} B_i^{(\beta)} - \sum_{i=1}^{\eta_T^{\gamma \to \alpha} + \eta_T^{\gamma \to \beta}} B_i^{(\gamma)}. \tag{86}$$

From group's perspective, the number of possible combinations of users within a group should be preserved because the size of each group is preserved. For the same reason, the number of possible combinations of users in distinct clusters are also conserved from cluster's viewpoint. These are reflected as:

$$\eta_T^{\alpha \to \beta} + \eta_T^{\alpha \to \gamma} = \eta_T^{\gamma \to \alpha} + \eta_T^{\beta \to \alpha}, \tag{87}$$

$$\eta_T^{\gamma \to \alpha} + \eta_T^{\gamma \to \beta} = \eta_T^{\alpha \to \gamma} + \eta_T^{\beta \to \gamma}. \tag{88}$$

In the case of $x < y$, where $x, y \in \{\alpha, \beta, \gamma\}$, $\eta_T^{x \to y}$ can be interpreted as the outgoing flow of edges from groups and clusters; otherwise, it can be interpreted as the ingoing flow of edges to groups and clusters. Then, due to the preservation law of total number of edges,

$$\eta_T^{\alpha \to \beta} + \eta_T^{\beta \to \gamma} + \eta_T^{\alpha \to \gamma} = \eta_T^{\beta \to \alpha} + \eta_T^{\gamma \to \beta} + \eta_T^{\gamma \to \alpha}. \tag{89}$$

Thus, by (87), (88) and (89), the RHS of (85) can be rewritten as

$$\sum_{j=1}^{\eta_T^{\alpha \to \beta}} \left( B_j^{(\alpha)} - B_j^{(\beta)} \right) + \sum_{\ell=1}^{\eta_T^{\beta \to \gamma}} \left( B_\ell^{(\gamma)} - B_\ell^{(\beta)} \right), \tag{90}$$

and the RHS of (86) is given by

$$\sum_{k=1}^{\eta_T^{\alpha \to \gamma}} \left( B_k^{(\alpha)} - B_k^{(\gamma)} \right) + \sum_{\ell=1}^{\eta_T^{\beta \to \gamma}} \left( B_\ell^{(\beta)} - B_\ell^{(\gamma)} \right). \tag{91}$$

On the other hand, one can compute

$$\eta_T^{\alpha \to \beta} = \sum_{x \in \{A,B\}} \sum_{i \in [3]} \left[ \underbrace{\left( \frac{n}{6} - \sum_{y \in \{A,B\}} \sum_{j \in [3]} k_{ij}^{xy} \right)}_{\text{Term}_1} \underbrace{\sum_{j \in [3]} k_{ij}^{xx}}_{\text{Term2}} \right], \tag{92}$$

$$\eta_T^{\alpha \to \gamma} = \sum_{x \in \{A,B\}} \sum_{i \in [3]} \left[ \left( \frac{n}{6} - \sum_{y \in \{A,B\} \setminus \{x\}} \sum_{j \in [3]} k_{ij}^{xy} \right) \sum_{y \in \{A,B\} \setminus \{x\}} \sum_{j \in [3]} k_{ij}^{xy} \right], \tag{93}$$

$$\eta_T^{\beta \to \gamma} = \sum_{x \in \{A,B\}} \sum_{i \in [3]} \left[ \left( \frac{n}{6} - \sum_{y \in \{A,B\} \setminus \{x\}} \sum_{j \in [3]} k_{ij}^{xy} \right) \sum_{y \in \{A,B\} \setminus \{x\}} \sum_{h \in [3] \setminus \{i\}} \sum_{j \in [3]} k_{hj}^{xy} \right]. \tag{94}$$

In (92), Term1 means the number of remaining users in $G_i^x$, and Term2 means the number of users that moved to other groups within cluster $x$. Note that (93) and (94) can be interpreted in a similar manner.

Thus, by (80) and (90) – (94), we obtain

$$\mathbb{P}\left[\mathsf{L}\left(M_0\right) \geq \mathsf{L}(X)\right] = \mathbb{P}\left[B \geq 0\right]$$

where $B$ refers to the quantity defined earlier in the statement of Lemma 2. This completes the first part of the proof (16) in Lemma 2.

Now, we will prove (17). Let

$$U_i := c_1 \, B_i^{(p)} \left(2B_i^{(\theta)} - 1\right), \quad i \in [1:n_1], \tag{95}$$

$$W_j := c_2 \left(B_j^{(\beta)} - B_j^{(\alpha)}\right), \quad j \in [n_1 + 1 : n_1 + n_2], \tag{96}$$

$$Y_k := c_3 \left(B_k^{(\gamma)} - B_k^{(\alpha)}\right), \quad k \in [n_1 + n_2 + 1 : n_1 + n_2 + n_3], \tag{97}$$

$$Z_\ell := c_4 \left(B_\ell^{(\gamma)} - B_\ell^{(\beta)}\right), \quad \ell \in [n_1 + n_2 + n_3 + 1 : n_1 + n_2 + n_3 + n_4]. \tag{98}$$

By Chernoff bound [21],

$$\mathbb{P}\left[B > 0\right] \leq \min_{t>0} \mathbb{E}\left[e^{tB}\right] \leq \mathbb{E}\left[e^{\frac{1}{2}B}\right] = \mathbb{E}\left[e^{\frac{1}{2}U_i}\right]^{n_1} \mathbb{E}\left[e^{\frac{1}{2}W_j}\right]^{n_2} \mathbb{E}\left[e^{\frac{1}{2}Y_k}\right]^{n_3} \mathbb{E}\left[e^{\frac{1}{2}Z_\ell}\right]^{n_4}. \tag{99}$$

We will calculate only $\mathbb{E}\left[e^{\frac{1}{2}U_i}\right]$ and $\mathbb{E}\left[e^{\frac{1}{2}W_j}\right]$, since $\mathbb{E}\left[e^{\frac{1}{2}Y_k}\right]$ and $\mathbb{E}\left[e^{\frac{1}{2}Z_\ell}\right]$ can be calculated in a similar way. One can evaluate $\mathbb{E}\left[e^{\frac{1}{2}U_i}\right]$ and $\mathbb{E}\left[e^{\frac{1}{2}W_j}\right]$ as follows

$$
\begin{aligned}
\mathbb{E}\left[e^{\frac{1}{2}U_i}\right] &= 1 - p + p\theta \exp\left(\frac{1}{2}\log\left(\frac{1-\theta}{\theta}\right)\right) + p(1-\theta)\exp\left(-\frac{1}{2}\log\left(\frac{1-\theta}{\theta}\right)\right) \\
&= 1 - p + 2p\sqrt{\theta(1-\theta)} \\
&= 1 - p\left(\sqrt{1-\theta} - \sqrt{\theta}\right)^2,
\end{aligned} \tag{100}
$$

$$
\begin{aligned}
\mathbb{E}\left[e^{\frac{1}{2}W_j}\right] &= (1-\alpha)(1-\beta) + \alpha\beta + (1-\alpha)\beta \exp\left(\frac{1}{2}\log\left(\frac{(1-\beta)\alpha}{(1-\alpha)\beta}\right)\right) \\
&\quad + \alpha(1-\beta)\exp\left(-\frac{1}{2}\log\left(\frac{(1-\beta)\alpha}{(1-\alpha)\beta}\right)\right) \\
&= (1-\alpha)(1-\beta) + \alpha\beta + 2\sqrt{\alpha\beta(1-\alpha)(1-\beta)} \\
&= \left(\sqrt{\alpha\beta} + \sqrt{(1-\alpha)(1-\beta)}\right)^2.
\end{aligned} \tag{101}
$$

Taking a negative log on both sides, we get

$$
\begin{aligned}
-\log\mathbb{E}\left[e^{\frac{1}{2}U_i}\right] &= -\log\left(1 - p\left(\sqrt{1-\theta} - \sqrt{\theta}\right)^2\right) \\
&= p\left(\sqrt{1-\theta} - \sqrt{\theta}\right)^2 + O\left(p^2\right) \tag{102} \\
&= (1 + o(1))\, I_r, \tag{103} \\
-\log\mathbb{E}\left[e^{\frac{1}{2}W_j}\right] &= -2\log\left(\sqrt{\alpha\beta} + \sqrt{(1-\alpha)(1-\beta)}\right) \\
&= -2\log\left(\sqrt{\alpha\beta} + \left(1 - \frac{1}{2}\alpha + O(\alpha^2)\right)\left(1 - \frac{1}{2}\beta + O(\beta^2)\right)\right) \tag{104} \\
&= -2\log\left(1 - \frac{1}{2}\alpha - \frac{1}{2}\beta + \sqrt{\alpha\beta} + O(\alpha^2 + \beta^2)\right) \\
&= \alpha + \beta - 2\sqrt{\alpha\beta} + O\left(\alpha^2 + \beta^2\right) \tag{105}
\end{aligned}
$$

$$= \left(\sqrt{\alpha} - \sqrt{\beta}\right)^2 + O\left(\alpha^2 + \beta^2\right)$$
$$= (1 + o(1)) I_g, \tag{106}$$

where (102) and (105) hold since $\log(1 + x) = x + O(x^2)$ for $x \simeq 0$; and (104) is due to $\sqrt{1 - x} = 1 - \frac{1}{2}x + O(x^2)$. Similarly, we obtain

$$-\log \mathbb{E}\left[e^{\frac{1}{2}Y_k}\right] = -\log\left(\sqrt{\alpha\gamma} + \sqrt{(1-\alpha)(1-\gamma)}\right)^2 = (1 + o(1))I_{c1}, \tag{107}$$

$$-\log \mathbb{E}\left[e^{\frac{1}{2}Z_\ell}\right] = -\log\left(\sqrt{\beta\gamma} + \sqrt{(1-\beta)(1-\gamma)}\right)^2 = (1 + o(1))I_{c2}. \tag{108}$$

Thus, we have

$$\mathbb{P}\left[B > 0\right] \leq \mathbb{E}\left[e^{\frac{1}{2}U_i}\right]^{n_1} \mathbb{E}\left[e^{\frac{1}{2}W_j}\right]^{n_2} \mathbb{E}\left[e^{\frac{1}{2}Y_k}\right]^{n_3} \mathbb{E}\left[e^{\frac{1}{2}Z_\ell}\right]^{n_4}$$
$$= \exp\left(n_1 \log \mathbb{E}\left[e^{\frac{1}{2}U_i}\right] + n_2 \log \mathbb{E}\left[e^{\frac{1}{2}W_j}\right] + n_3 \log \mathbb{E}\left[e^{\frac{1}{2}Y_k}\right] + n_4 \log \mathbb{E}\left[e^{\frac{1}{2}Z_\ell}\right]\right)$$
$$= \exp\left(-(1 + o(1))(n_1 I_r + n_2 I_g + n_3 I_{c1} + n_4 I_{c2})\right). \tag{109}$$

This completes the proof of (17), and concludes the proof of Lemma 2. $\qquad\square$

## A.3 Proof of Lemma 3

First, we show that if either $k_{ij}^{xy} > \frac{\tau}{5}n$ or $d_i^x(\ell) > \tau m$ holds for some $(x, y, i, j, \ell)$, then $\Lambda(M_0, X_T) = \Omega(nm)$ holds. Suppose there exists $x, y, i^\star, j^\star$ such that $k_{i^\star j^\star}^{xy} > \frac{\tau}{5}n$, $k_{ij}^{xy} \leq \frac{\tau}{5}n$ for $i \in [3] \setminus \{i^\star\}, j \in [3] \setminus \{j^\star\}$ and $d_i^x(\ell) \leq \tau m$. Then, the following inequality holds from (84)

$$\Lambda(M_0, X)$$

$$\geq \sum_{x \in \{A,B\}} \sum_{i \in [3]} \sum_{y \in \{A,B\}} \sum_{j \in [3]} k_{ij}^{xy} \left( d_H\left(v_i^x, v_j^y\right) - \sum_{\ell \in \Delta(v_i^x, v_j^y)} d_i^x(\ell) + \sum_{\ell \in \{0,1\}^3 \setminus \Delta(v_i^x, v_j^y)} d_i^x(\ell) \right)$$

$$\geq \frac{\tau}{5}n \left( \min\{\delta_g, \delta_c\} - \max_{\substack{i,j \in [3] \\ x,y \in \{A,B\}}} |\Delta(v_i^x, v_j^x)| \cdot \tau \right) m = \Omega(nm). \tag{110}$$

This is because

$$d_H\left(v_i^x, v_j^y\right) \geq m \cdot \min\{\delta_g, \delta_c\},$$

and

$$\sum_{\ell \in \Delta(v_i^x, v_j^y)} d_i^x(\ell) \leq \tau m \cdot \max_{\substack{i,j \in [3] \\ x,y \in \{A,B\}}} |\Delta(v_i^x, v_j^x)|. \tag{111}$$

Suppose there exists $x, i^\star, \ell^\star$ such that $d_{i^\star, \ell^\star}^x > \tau m$, $d_{i,\ell}^x < \tau m$ for all $i \in [3] \setminus \{i^\star\}, \ell \in \{0,1\}^3 \setminus \{\ell^\star\}$ and $k_{ij}^{xy} \leq \frac{\tau}{5}n$. Since $\sum_{j \in [3], y \in \{A,B\}} k_{ij}^{xy} \leq \frac{5}{36}n$, the following inequality holds from (84)

$$\Lambda(M_0, X) \geq \sum_{x \in \{A,B\}} \sum_{i \in [3]} \left( \frac{n}{6} - \sum_{y \in \{A,B\}} \sum_{j \in [3]} k_{ij}^{xy} \right) \left( \sum_{\ell \in \{0,1\}^3} d_i^x(\ell) \right) \geq \frac{1}{36}n \cdot \tau m = \Omega(nm). \tag{112}$$

Also, $\sum_{T \in \mathcal{T} \setminus \mathcal{T}_1} |\mathcal{X}(T)| \leq 6^n 2^{6m}$ holds. Here $6^n 2^{6m}$ represents the total number of possible configurations of rating matrices. Since (5), (6) and (7) imply $mnI_r = \Omega(n \log n + m \log m)$, the first term of (19) is upper bounded by

$$\sum_{T \in \mathcal{T}^{(\delta)} \setminus \mathcal{T}_1^{(\delta)}} |\mathcal{X}(T)| \exp\left(-(1 + o(1))\Lambda(M_0, X)I_r\right) \leq 6^n 2^{6m} e^{-\Omega(nm)I_r} \leq \left(\frac{6}{n}\right)^n \left(\frac{2^6}{m}\right)^m.$$

This completes the proof of Lemma 3. $\qquad\square$

## A.4 Proof of Lemma 4

We calculate the upper bound on the cardinality of matrix class $|\mathcal{X}(T)|$ as

$$|\mathcal{X}(T)| = |\{k_{ij}^{xy}\}_{i,j\in[3],x,y\in\{A,B\}}| \, |\{d_i^x(\ell)\}_{i\in[3],\ell\in\{0,1\}^3,x\in\{A,B\}}|.$$

The following inequality holds

$$|\{k_{ij}^{xy}\}_{i,j\in[3],x,y\in\{A,B\}}| \le \prod_{\substack{x,y\in\{A,B\}\\i,j\in[3]}} \binom{\frac{n}{6}}{k_{ij}^{xy}}, \tag{113}$$

since we choose $\{k_{ij}^{xy}\}_{j\in[3],y\in\{A,B\}}$ users from $i^{\text{th}}$ group of size $\frac{n}{6}$ in a cluster $x$.

Next, $|\{d_i^x(\ell)\}_{i\in[3],\ell\in\{0,1\}^3,x\in\{A,B\}}|$ is equal to the number of cases where we first choose $d_1^x(\ell)$ columns in $v_1^x$ and $(d_2^x(\ell) - d_{\text{ov}}^x(\ell))$ columns in $v_2^x$ among $m\delta_\ell$ columns, and then choose $d_{\text{ov}}^x(\ell)$ columns among $d_1^x(\ell)$ columns within the column block $I_\ell$ for $x \in \{A,B\}$. Thus, $|\{d_i^x(\ell)\}_{i\in[3],\ell\in\{0,1\}^3,x\in\{A,B\}}|$ is equal to

$$\prod_{\substack{x\in\{A,B\}\\\ell\in\{0,1\}^3}} \binom{m\delta_\ell}{d_1^x(\ell), d_2^x(\ell) - d_{\text{ov}}^x(\ell)}\binom{d_1^x(\ell)}{d_{\text{ov}}^x(\ell)}. \tag{114}$$

By (113) and (114), the following holds

$$|\mathcal{X}(T)| \le \prod_{\substack{x\in\{A,B\}\\\ell\in\{0,1\}^3}} m\binom{\delta_\ell}{d_1^x(\ell), d_2^x(\ell) - d_{\text{ov}}^x(\ell)}\binom{d_1^x(\ell)}{d_{\text{ov}}^x(\ell)} \prod_{\substack{x,y\in\{A,B\}\\i,j\in[3]}} \binom{\frac{n}{6}}{k_{ij}^{xy}}$$

$$\le \prod_{\substack{x\in\{A,B\}\\\ell\in\{0,1\}^3}} \exp\left((d_1^x(\ell) + d_2^x(\ell) - d_{\text{ov}}^x(\ell))\log m + d_1^x(\ell)\right) \prod_{\substack{x,y\in\{A,B\}\\i,j\in[3]}} n^{k_{ij}^{xy}} \tag{115}$$

$$= \exp\left(\left(\sum_{x\in\{A,B\}}\sum_{\ell\in\{0,1\}^3} d_1^x(\ell) + d_2^x(\ell) - d_{\text{ov}}^x(\ell)\right)\log m + \sum_{x\in\{A,B\}}\sum_{\ell\in\{0,1\}^3} d_1^x(\ell)\right)$$

$$\times \exp\left(\left(\sum_{x\in\{A,B\}}\sum_{i\in[3]}\sum_{y\in\{A,B\}}\sum_{j\in[3]} k_{ij}^{xy}\right)\log n\right) \tag{116}$$

$$= \exp\left(\frac{d_t}{2}\log m + \sum_{x\in\{A,B\}}\sum_{\ell\in\{0,1\}^3} d_1^x(\ell) + (k_g + k_c)\log n\right), \tag{117}$$

where (115) follows by $\binom{a}{b} \le a^b = \exp(b\log a)$ and $\binom{m}{n} \le 2^m \le e^m$.

Under the conditions of $k_{ij}^{xy} \le \frac{\tau}{5}n$ and $d_i^x(\ell) \le \tau m$, the following inequalities hold from (84), (92), (93) and (94)

$$\eta_T^{\alpha\to\beta} \ge \left(\frac{1}{6} - \tau\right)nk_g, \tag{118}$$

$$\eta_T^{\alpha\to\gamma} \ge \left(\frac{1}{6} - \frac{3}{5}\tau\right)nk_c, \tag{119}$$

$$\eta_T^{\beta\to\gamma} \ge \left(\frac{1}{3} - \frac{6}{5}\tau\right)nk_c, \tag{120}$$

$$\Lambda(M_0, X_T) \ge (\delta_g - \tau_g)k_g + (\delta_c - \tau_c)k_c + \left(\frac{1}{6} - \tau\right)nd_{\text{total}}, \tag{121}$$

where

$$\tau_g := \max_{\substack{i,j\in[3]\\x,y\in\{A,B\}}} |\Delta(v_i^x, v_j^x)| \cdot \tau,$$

$$\tau_c := \max_{\substack{i,j \in [3] \\ x \neq y}} |\Delta(v_i^x, v_j^y)| \cdot \tau.$$

By (117) – (121), the second term of (19) is upper bounded by

$$\sum_{T \in \mathcal{T}_1^{(\delta)}} \exp\left(-\frac{C_1}{2} d_t - C_2 k_g - C_3 k_c\right) \tag{122}$$

where

$$C_1 := \left(\frac{1}{3} - 2\tau\right) n I_r - \log m - 1, \tag{123}$$

$$C_2 := (\delta_g - \tau_g) m I_r + \left(\frac{1}{6} - \tau\right) n I_g - \log n, \tag{124}$$

$$C_3 := (\delta_c - \tau_c) m I_r + \left(\frac{1}{6} - \frac{3}{5}\tau\right) n I_{c1} + \left(\frac{1}{3} - \frac{6}{5}\tau\right) n I_{c2} - \log n. \tag{125}$$

For sufficiently large $n$ and $m$, the following inequalities hold from (5), (6) and (7)

$$C_1 \geq \frac{\epsilon}{2} \log m, \tag{126}$$

$$C_2 \geq \frac{\epsilon}{2} \log n, \tag{127}$$

$$C_3 \geq \frac{\epsilon}{2} \log n. \tag{128}$$

Thus, by (126) –(128), (122) is upper bounded by

$$\sum_{T \in \mathcal{T}_1^{(\delta)}} \exp\left(-\frac{\epsilon}{4} d_t \log m - \frac{\epsilon}{2}(k_g + k_c) \log n\right). \tag{129}$$

Now, we show that (129) converges to 0 as $n$ and $m$ tend to infinity. Let $k_t := k_g + k_c$. Note that $k_t$ is an even number because

$$\sum_{y \in \{A,B\}} \sum_{j \in [3]} k_{ij}^{xy} = \sum_{y \in \{A,B\}} \sum_{j \in [3]} k_{ji}^{yx}$$

holds for all $i$ and $x$ since the size of group should be preserved. Also, $d_t$ is an even number due to

$$d_t = \sum_{x \in \{A,B\}} \sum_{\ell \in \{0,1\}^3} d_1^x(\ell) + d_2^x(\ell) + d_3^x(\ell)$$

$$= \sum_{x \in \{A,B\}} \sum_{\ell \in \{0,1\}^3} 2\left(d_1^x(\ell) + d_2^x(\ell) - d_{\text{ov}}^x(\ell)\right).$$

The maximum value of $k_t$ is $6\tau m$ and $d_t$ is $48\tau m$ by the definition of $\mathcal{T}_1^{(\delta)}$. Then, (129) is upper bounded by

$$\sum_{T \in \mathcal{T}_1^{(\delta)}} \exp\left(-\frac{\epsilon}{4} d_t \log m - \frac{\epsilon}{2} k_t \log n\right)$$

$$\leq n^{-\epsilon} + m^{-\frac{\epsilon}{2}} + \sum_{k=2}^{6\tau n} \sum_{d=2}^{48\tau m} n^{-\frac{\epsilon}{2}k} m^{-\frac{\epsilon}{4}d} \cdot |\{T : k_t = k\}| \cdot |\{T : d_t = d\}|$$

$$\leq n^{-\epsilon} + m^{-\frac{\epsilon}{2}} + \sum_{k=2}^{6\tau n} \sum_{d=2}^{48\tau m} n^{-\frac{\epsilon}{2}k} m^{-\frac{\epsilon}{4}d} \cdot \binom{k+29}{29} \cdot \binom{d+47}{47} \tag{130}$$

$$\leq n^{-\epsilon} + m^{-\frac{\epsilon}{2}} + 2^{76} \sum_{k=2}^{6\tau n} \sum_{d=2}^{48\tau m} (2n^{-\frac{\epsilon}{2}})^k (2m^{-\frac{\epsilon}{4}})^d \tag{131}$$

$$= n^{-\epsilon} + m^{-\frac{\epsilon}{2}} + 2^{76} \cdot \frac{1 - (4n^{-\epsilon})^{3\tau n}}{1 - 4n^{-\epsilon}} \cdot \frac{1 - (4m^{-\frac{\epsilon}{2}})^{24\tau m}}{1 - 4m^{-\frac{\epsilon}{2}}} \cdot 4n^{-\epsilon} \cdot 4m^{-\frac{\epsilon}{2}}, \tag{132}$$

where (130) follows from the fact that the number of cases of $\sum_{i=1}^{n} x_i = r, x_i \geq 0$ for all $i$ is equal to $\binom{r+n-1}{n-1}$; and (131) is due to $\binom{a}{b} \leq 2^a$. Since (132) goes to zero as $n, m \to \infty$, this completes the proof of Lemma 4. $\qquad \square$

# B  Proofs of Lemmas for Converse Proof of Theorem 1

## B.1  Proof of Lemma 5

We will follow a similar proof technique to that of Lemma 5.2 in [22]. Recall that we denote by $B^{(\mu)}$ a Bernoulli random variable with parameter $\mu$, that is, $\mathbb{P}[B^{(\mu)} = 1] = 1 - \mathbb{P}[B^{(\mu)} = 0] = \mu$.

For $p = \Theta\left(\frac{\log n}{n}\right)$ and a constant $\theta \in [0,1]$, we can define $X(p,\theta) = \log\left(\frac{1-\theta}{\theta}\right) B^{(p)}(2B^{(\theta)} - 1)$, with $c' = \log\left(\frac{1-\theta}{\theta}\right)$, that is,

$$X(p,\theta) = \begin{cases} -\log\left(\frac{1-\theta}{\theta}\right) & \text{w.p. } p(1-\theta), \\ 0 & \text{w.p. } 1-p, \\ \log\left(\frac{1-\theta}{\theta}\right) & \text{w.p. } p\theta. \end{cases}$$

Then, we can evaluate the moment generating function of $X(p,\theta)$ at $t = 1/2$ as

$$M_{X(p,\theta)}\left(\frac{1}{2}\right) = \mathbb{E}\left[\exp(X/2)\right]$$

$$= p(1-\theta)\exp\left(-\frac{1}{2}\log\left(\frac{1-\theta}{\theta}\right)\right) + (1-p) + p\theta\exp\left(\frac{1}{2}\log\left(\frac{1-\theta}{\theta}\right)\right)$$

$$= p(1-\theta)\sqrt{\frac{\theta}{1-\theta}} + (1-p) + p\theta\sqrt{\frac{1-\theta}{\theta}}$$

$$= 2p\sqrt{\theta(1-\theta)} + 1 - p, \tag{133}$$

which implies

$$-\log M_{X(p,\theta)}\left(\frac{1}{2}\right) = (1 + o(1))(\sqrt{1-\theta} - \sqrt{\theta})^2 p. \tag{134}$$

We also define $\widehat{X} = \widehat{X}(p,\theta)$ as a new random variable with the same range as $X(p,\theta)$, and probability mass function given by

$$f_{\widehat{X}}(x) = \frac{\exp(\frac{x}{2})f_X(x)}{M_X(\frac{1}{2})}.$$

More precisely, we have

$$\widehat{X}(p,\theta) = \begin{cases} -\log\left(\frac{1-\theta}{\theta}\right) & \text{w.p. } \frac{p\sqrt{\theta(1-\theta)}}{M_X(\frac{1}{2})}, \\ 0 & \text{w.p. } \frac{1-p}{M_X(\frac{1}{2})}, \\ \log\left(\frac{1-\theta}{\theta}\right) & \text{w.p. } \frac{p\sqrt{\theta(1-\theta)}}{M_X(\frac{1}{2})}. \end{cases}$$

Then it is straightforward to see that

$$\mathbb{E}[\widehat{X}(p,\theta)] = 0 \tag{135}$$

$$\mathsf{Var}[\widehat{X}(p,\theta)] = \frac{2p\sqrt{\nu(1-\theta)}}{2p\sqrt{\theta(1-\theta)} + 1 - p}\left(\log\frac{1-\theta}{\theta}\right)^2 = O(p). \tag{136}$$

Next, for $\mu, \nu = \Theta\left(\frac{\log n}{n}\right) [0,1]$, define $Y(\mu,\nu) = c(B^{(\mu)} - B^{(\nu)})$, where $c = \log\left(\frac{(1-\mu)\nu}{(1-\nu)\mu}\right)$. More precisely, we have

$$Y(\mu,\nu) = \begin{cases} -\log\left(\frac{(1-\mu)\nu}{(1-\nu)\mu}\right) & \text{w.p. } (1-\mu)\nu, \\ 0 & \text{w.p. } (1-\mu)(1-\nu) + \mu\nu, \\ \log\left(\frac{(1-\mu)\nu}{(1-\nu)\mu}\right) & \text{w.p. } \mu(1-\nu). \end{cases}$$

The moment generating function of $Y(\mu,\nu)$ at $t = 1/2$ is given by

$$M_{Y(\mu,\nu)}\left(\frac{1}{2}\right) = \mathbb{E}[\exp(Y/2)]$$

$$= (1-\mu)\nu \exp(-c/2) + \mu(1-\nu)\exp(c/2) + (1-\mu)(1-\nu) + \mu\nu$$

$$= (1-\mu)\nu\sqrt{\frac{(1-\nu)\mu}{(1-\mu)\nu}} + (1-\nu)\mu\sqrt{\frac{(1-\mu)\nu}{(1-\nu)\mu}} + (1-\mu)(1-\nu) + \mu\nu$$

$$= 2\sqrt{(1-\mu)(1-\nu)\mu\nu} + (1-\mu)(1-\nu) + \mu\nu$$

$$= \left(\sqrt{\mu\nu} + \sqrt{(1-\mu)(1-\nu)}\right)^2, \tag{137}$$

which implies

$$-\log M_{Y(\mu,\nu)}\left(\frac{1}{2}\right) = (1+o(1))\left(\sqrt{\nu} - \sqrt{\mu}\right)^2. \tag{138}$$

Define a random variable $\widehat{Y} = \widehat{Y}(\mu,\nu)$ with $f_{\widehat{Y}}(y) = \frac{\exp(\frac{y}{2})f_Y(y)}{M_Y(\frac{1}{2})}$. Then, for $\widehat{Y}(\mu,\nu)$, we have

$$\mathbb{E}[\widehat{Y}(\mu,\nu)] = \frac{1}{M_Y(\frac{1}{2})}\left[-(1-\mu)\nu\exp\left(-\frac{c}{2}\right)\cdot c + \mu(1-\nu)\exp\left(\frac{c}{2}\right)\cdot c\right]$$

$$= \frac{1}{M_Y(\frac{1}{2})}\left[-(1-\mu)\nu\sqrt{\frac{(1-\nu)\mu}{(1-\mu)\nu}}\cdot c + (1-\nu)\mu\sqrt{\frac{(1-\mu)\nu}{(1-\nu)\mu}}\cdot c\right]$$

$$= \frac{1}{M_Y(\frac{1}{2})}\left[-\sqrt{(1-\mu)(1-\nu)\mu\nu}\cdot c + \sqrt{(1-\mu)(1-\nu)\mu\nu}\cdot c\right] = 0, \tag{139}$$

and

$$\mathsf{Var}[\widehat{Y}(\mu,\nu)] = \frac{\sqrt{(1-\mu)(1-\nu)\mu\nu}}{\left(\sqrt{\mu\nu} + \sqrt{(1-\mu)(1-\nu)}\right)^2}\left(\log\frac{(1-\mu)\nu}{(1-\nu)\mu}\right)^2 = O\left(\sqrt{\mu\nu}\right), \tag{140}$$

where $\mu, \nu = \Theta\left(\frac{\log n}{n}\right)$.

Now, we can rewrite the random variable of interest in the lemma as

$$B := \sum_{i=1}^{n_1}\log\left(\frac{1-\theta}{\theta}\right)B_i^{(p)}\left(2B_i^{(\theta)}-1\right) + \sum_{j=n_1+1}^{n_1+n_2}\log\left(\frac{(1-\beta)\alpha}{(1-\alpha)\beta}\right)\left(B_j^{(\beta)}-B_j^{(\alpha)}\right)$$

$$+ \sum_{k=n_1+n_2+1}^{n_1+n_2+n_3}\log\left(\frac{(1-\gamma)\alpha}{(1-\alpha)\gamma}\right)\left(B_k^{(\gamma)}-B_k^{(\alpha)}\right) + \sum_{\ell=n_1+n_2+n_3+1}^{n_1+n_2+n_3+n_4}\log\left(\frac{(1-\gamma)\beta}{(1-\beta)\gamma}\right)\left(B_\ell^{(\gamma)}-B_\ell^{(\beta)}\right),$$

$$= \sum_{i=1}^{n_1}X_i(p,\theta) + \sum_{j=n_1+1}^{n_1+n_2}Y_j(\beta,\alpha) + \sum_{k=n_1+n_2+1}^{n_1+n_2+n_3}Y_k(\gamma,\alpha) + \sum_{\ell=n_1+n_2+n_3+1}^{n_1+n_2+n_3+n_4}Y_\ell(\gamma,\beta). \tag{141}$$

Therefore, we can write

$$\mathbb{P}\left[B \geq 0\right]$$

$$= \mathbb{P}\left[\sum_{i=1}^{n_1}X_i(p,\theta) + \sum_{j=n_1+1}^{n_1+n_2}Y_j(\beta,\alpha) + \sum_{k=n_1+n_2+1}^{n_1+n_2+n_3}Y_k(\gamma,\alpha) + \sum_{\ell=n_1+n_2+n_3+1}^{n_1+n_2+n_3+n_4}Y_\ell(\gamma,\beta) \geq 0\right]$$

$$\geq \mathbb{P}\left[0 \leq \sum_{i=1}^{n_1}X_i(p,\theta) + \sum_{j=n_1+1}^{n_1+n_2}Y_j(\beta,\alpha) + \sum_{k=n_1+n_2+1}^{n_1+n_2+n_3}Y_k(\gamma,\alpha) + \sum_{\ell=n_1+n_2+n_3+1}^{n_1+n_2+n_3+n_4}Y_\ell(\gamma,\beta) < \xi\right]$$

$$\overset{(a)}{=} \sum_{\mathcal{R}(\xi)}\left[\prod_{i=1}^{n_1}f_{X(p,\theta)}(x_i)\prod_{j=n_1+1}^{n_1+n_2}f_{Y(\beta,\alpha)}(y_j)\prod_{k=n_1+n_2+1}^{n_1+n_2+n_3}f_{Y(\gamma,\alpha)}(y_k)\prod_{\ell=n_1+n_2+n_3+1}^{n_1+n_2+n_3+n_4}f_{Y(\gamma,\beta)}(y_\ell)\right]$$

$$\overset{(b)}{\geq} \frac{\left(M_{X(p,\theta)}\left(\frac{1}{2}\right)\right)^{n_1}\left(M_{Y(\beta,\alpha)}\left(\frac{1}{2}\right)\right)^{n_2}\left(M_{Y(\gamma,\alpha)}\left(\frac{1}{2}\right)\right)^{n_3}\left(M_{Y(\gamma,\beta)}\left(\frac{1}{2}\right)\right)^{n_4}}{\exp\left(\frac{1}{2}\xi\right)}$$

$$\times \sum_{\mathcal{R}(\xi)} \left[ \prod_{i=1}^{n_1} \frac{\exp\left(\frac{1}{2}x_i\right) f_{X(p,\theta)}(x_i)}{M_{X(p,\theta)}\left(\frac{1}{2}\right)} \prod_{j=n_1+1}^{n_1+n_2} \frac{\exp\left(\frac{1}{2}y_j\right) f_{Y(\beta,\alpha)}(y_j)}{M_{Y(\beta,\alpha)}\left(\frac{1}{2}\right)} \right.$$

$$\left. \cdot \prod_{k=n_1+n_2+1}^{n_1+n_2+n_3} \frac{\exp\left(\frac{1}{2}y_k\right) f_{Y(\gamma,\alpha)}(y_k)}{M_{Y(\gamma,\alpha)}\left(\frac{1}{2}\right)} \prod_{\ell=n_1+n_2+n_3+1}^{n_1+n_2+n_3+n_4} \frac{\exp\left(\frac{1}{2}y_\ell\right) f_{Y(\gamma,\beta)}(y_\ell)}{M_{Y(\gamma,\beta)}\left(\frac{1}{2}\right)} \right]$$

$$= \exp\left( n_1 \log M_{X(p,\theta)}\left(\frac{1}{2}\right) + n_2 \log M_{Y(\beta,\alpha)}\left(\frac{1}{2}\right) + n_3 \log M_{Y(\gamma,\alpha)}\left(\frac{1}{2}\right) + n_4 \log M_{Y(\gamma,\beta)}\left(\frac{1}{2}\right) - \frac{1}{2}\xi \right)$$

$$\times \sum_{\mathcal{R}(\xi)} \left[ \prod_{i=1}^{n_1} f_{\widehat{X}(p,\theta)}(x_i) \prod_{j=n_1+1}^{n_1+n_2} f_{\widehat{Y}(\beta,\alpha)}(y_j) \prod_{k=n_1+n_2+1}^{n_1+n_2+n_3} f_{\widehat{Y}(\gamma,\alpha)}(y_k) \prod_{\ell=n_1+n_2+n_3+1}^{n_1+n_2+n_3+n_4} f_{\widehat{Y}(\gamma,\beta)}(y_\ell) \right]$$

$$\stackrel{(c)}{=} \exp\left( -(1+o(1))(n_1 I_r + n_2 I_g + n_3 I_{c1} + n_4 I_{c2}) - \frac{1}{2}\xi \right)$$

$$\times \mathbb{P}\left[ 0 \le \sum_{i=1}^{n_1} \widehat{X}_i(p,\theta) + \sum_{j=n_1+1}^{n_1+n_2} \widehat{Y}_j(\beta,\alpha) + \sum_{k=n_1+n_2+1}^{n_1+n_2+n_3} \widehat{Y}_k(\gamma,\alpha) + \sum_{\ell=n_1+n_2+n_3+1}^{n_1+n_2+n_3+n_4} \widehat{Y}_\ell(\gamma,\beta) < \xi \right],$$

$$(142)$$

where $(a)$ follows from independence of $X_i(\cdot,\cdot)$'s and $Y_i(\cdot,\cdot)$'s variables in (141) since their indices are different, hence they are generated from independent Bernoulli random variables, and note that the summation in $(a)$ is over

$$\mathcal{R}(\xi) = \left\{ \{x_i\}_{i=1}^{n_1}, \{y_j\}_{j=n_1+1}^{n_1+n_2+n_3+n_4} : 0 \le \sum_{i=1}^{n_1} x_i + \sum_{j=n_1+1}^{n_1+n_2+n_3+n_4} y_j < \xi \right\}.$$

Moreover, $(b)$ holds since $\exp\left( \frac{1}{2}\left( \sum_{i=1}^{n_1} x_i + \sum_{j=n_1+1}^{n_1+n_2+n_3+n_4} y_j \right) \right) < \exp\left(\frac{1}{2}\xi\right)$; and $(c)$ holds due to the independence of $\widehat{Y}_i(\cdot,\cdot)$'s and $\widehat{X}_i(\cdot,\cdot)$'s. Finally $I_r$, $I_g$, $I_{c1}$ and $I_{c2}$ in (142) are given by

$$I_r = p\left( \sqrt{1-\theta} - \sqrt{\theta} \right)^2,$$

$$I_g = \left( \sqrt{\alpha} - \sqrt{\beta} \right)^2,$$

$$I_{c1} = \left( \sqrt{\alpha} - \sqrt{\gamma} \right)^2,$$

$$I_{c2} = \left( \sqrt{\beta} - \sqrt{\gamma} \right)^2,$$

which follow from (134) and (138).

Note that (142) holds for any value of $\xi$. In particular, we can choose $\xi_n$ satisfying

$$\lim_{n\to\infty} \frac{\xi_n}{n_1 I_r + n_2 I_g + n_3 I_{c1} + n_4 I_{c2}} = 0, \tag{143}$$

$$\lim_{n\to\infty} \frac{n_1 p + n_2 \sqrt{\alpha\beta} + n_3 \sqrt{\alpha\gamma} + n_4 \sqrt{\beta\gamma}}{\xi_n^2} = 0. \tag{144}$$

Therefore, (143) implies that the exponent in (142) can be rewritten as

$$-(1+o(1))\left( n_1 I_r + n_2 I_g + n_3 I_{c1} + n_4 I_{c2} \right) - \frac{1}{2}\xi_n = -(1+o(1))\left( n_1 I_r + n_2 I_g + n_3 I_{c1} + n_4 I_{c2} \right). \tag{145}$$

Moreover, the probability in (142) can be bounded as

$$\mathbb{P}\left[ 0 \le \sum_{i=1}^{n_1} \widehat{X}_i(p,\theta) + \sum_{j=n_1+1}^{n_1+n_2} \widehat{Y}_j(\beta,\alpha) + \sum_{k=n_1+n_2+1}^{n_1+n_2+n_3} \widehat{Y}_k(\gamma,\alpha) + \sum_{\ell=n_1+n_2+n_3+1}^{n_1+n_2+n_3+n_4} \widehat{Y}_\ell(\gamma,\beta) < \xi_n \right]$$

$$\overset{(a)}{\geq} \frac{1}{2} - \mathbb{P}\left[ \sum_{i=1}^{n_1} \widehat{X}_i(p,\theta) + \sum_{j=n_1+1}^{n_1+n_2} \widehat{Y}_j(\beta,\alpha) + \sum_{k=n_1+n_2+1}^{n_1+n_2+n_3} \widehat{Y}_k(\gamma,\alpha) + \sum_{\ell=n_1+n_2+n_3+1}^{n_1+n_2+n_3+n_4} \widehat{Y}_\ell(\gamma,\beta) \geq \xi_n \right]$$

$$\overset{(b)}{\geq} \frac{1}{2} - \frac{n_1\mathsf{Var}[\widehat{X}(p,\theta)] + n_2\mathsf{Var}[\widehat{Y}(\beta,\alpha)] + n_3\mathsf{Var}[\widehat{Y}(\gamma,\alpha)] + n_4\mathsf{Var}[\widehat{Y}(\gamma,\beta)]}{\xi_n^2}$$

$$\overset{(c)}{=} \frac{1}{2} - \frac{n_1 O(p) + n_2 O(\sqrt{\alpha\beta}) + n_3 O(\sqrt{\alpha\gamma}) + n_4 O(\sqrt{\beta\gamma})}{\xi_n^2}$$

$$\overset{(d)}{=} \frac{1}{2} - o(1) > \frac{1}{4}, \tag{146}$$

where $(a)$ is due to the symmetry of random variables $\widehat{X}(\cdot,\cdot)$ and $\widehat{Y}(\cdot,\cdot)$, $(b)$ follows from Chebyshev's inequality, in $(c)$ the variances are replaced by (136) and (140), and finally $(d)$ is a consequence of (144). Plugging (145) and (146) in (142), we get the desired bound in Lemma 5. $\qquad\square$

## B.2  Proof of Lemma 6

The proof hinges on the alteration method [23]. We present a constructive proof for the existence of subgroups $\tilde{G}_1^A$ and $\tilde{G}_2^A$. Let $r = \frac{n}{\log^3 n}$. We start by sampling two random subsets $\overline{G}_i^A$ from $G_i^A$ of size $|\overline{G}_i^A| = 2r$, for $i = 1, 2$. Then, we prune these sets to obtain the desired edge free subsets. To this end, for any pair of nodes $f, g \in \overline{G}_1^A \cup \overline{G}_2^A$, we remove both $f$ and $g$ from $\overline{G}_1^A \cup \overline{G}_2^A$ if $(f, g) \in E$. We continue this process until the remaining set of nodes is edge-free. Let $\mathcal{P}$ be the set of nodes we remove from $\overline{G}_1^A \cup \overline{G}_2^A$ throughout the pruning process. The expected value of $\mathcal{P}$ can be upper bounded by

$$\mathbb{E}[|\mathcal{P}|] \leq 2\mathbb{E}\left[ \sum_{f,g \in \overline{G}_1^A \cup \overline{G}_2^A} \mathbb{1}\left[(f,g) \in E\right] \right]$$

$$= 2 \sum_{f,g \in \overline{G}_1^A} \mathbb{E}[\mathbb{1}\left[(f,g) \in E\right]] + 2 \sum_{f,g \in \overline{G}_2^A} \mathbb{E}[\mathbb{1}\left[(f,g) \in E\right]] + 2 \sum_{f \in \overline{G}_1^A} \sum_{g \in \overline{G}_2^A} \mathbb{E}[\mathbb{1}\left[(f,g) \in E\right]]$$

$$= 2 \sum_{f,g \in \overline{G}_1^A} \alpha + 2 \sum_{f,g \in \overline{G}_2^A} \alpha + 2 \sum_{f \in \overline{G}_1^A} \sum_{g \in \overline{G}_2^A} \beta$$

$$= 2\binom{2r}{2}\alpha + 2\binom{2r}{2}\alpha + 2(2r)^2\beta \leq 16r^2\alpha$$

where the last inequality holds since $\beta < \alpha$. Using Markov's inequality for the non-negative random variable $|\mathcal{P}|$, we obtain

$$\mathbb{P}\left[|\mathcal{P}| \geq r\right] \leq \frac{\mathbb{E}[N]}{r} \leq \frac{16n}{\log^3 n}\alpha = \Theta\left(\frac{n}{\log^3 n} \times \frac{\log n}{n}\right) = o(1). \tag{147}$$

Therefore, the number of remaining nodes (after pruning) satisfies

$$\mathbb{P}\left[|\overline{G}_1^A \cup \overline{G}_2^A \setminus \mathcal{P}| > 3r\right] = \mathbb{P}\left[|\mathcal{P}| < r\right] = 1 - \mathbb{P}\left[|\mathcal{P}| \geq r\right] = 1 - o(1).$$

Hence, $\overline{G}_1^A \setminus \mathcal{P}$ and $\overline{G}_2^A \setminus \mathcal{P}$ together have at least $3r$ elements. This, together with the fact that $|\overline{G}_1^A| = |\overline{G}_2^A| = 2r$, implies each of $\overline{G}_1^A \setminus \mathcal{P}$ and $\overline{G}_2^A \setminus \mathcal{P}$ have at least $r$ elements. Therefore, we can choose $r$ from $\overline{G}_i^A \setminus \mathcal{P}$ to form the desired set $\tilde{G}_i^A$, for $i = 1, 2$. This completes the proof of Lemma 6. $\qquad\square$

## Footnotes

[2]Exact recovery requires the number of wrongly clustered users vanishes as the number of users tends to infinity. The formal mathematical definition is given in [5, Definition 4].

[3]Almost exact recovery means that groups are recovered with a vanishing *fraction* of misclassified users. The mathematical definition is given in [5, Definition 4].