[Reviews · NeurIPS 2020]

Review 1

Summary and Contributions: The paper studies a binary version of the matrix completion problem, where we are additionally given a graph that provides "side information". Specifically, there is an ideal ratings matrix (users, items), and there is a graph between the users. The ideal matrix is assumed to have a _very stylized_ structure: users are divided into precisely two clusters of equal size, each cluster is divided into three groups of equal size, and within each group all the users have _exactly the same_ rating vector. (There's an additional linear algebraic dependence which I will ignore here.) The side information graph is assumed to be a block model whose parameters correspond to edge probability within groups, within cluster but across groups, and across clusters (and these are in decreasing order). In this setting, given a partially observed and independently corrupted ranking matrix, the paper gives an iterative refinement algorithm that can _perfectly_ reconstruct the ideal matrix with a good probability. They give conditions under which the error probability goes to 0.

Strengths: - The algorithm is fairly clean, and the fact that perfect recovery is possible is nice. - Taking advantage of side information in a provable way is a nice feature that is often difficult to prove.

Weaknesses: - The setting is a bit too stylized, and I found it very unconvincing. Perhaps the authors can generalize at least to a "k distinct vectors" and "t clusters" model instead of 2 vectors and 2 clusters. - The boolean operations are also unrealistic. Is there an intuitive rationale for one group of people having rating vector x, another y, and a third having x \XOR y ? - While there are technical improvements, the substantial improvement over [39, 40] seems unconvincing. - Experiments are primarily on synthetic data. The only real dataset appears to be one where the graph is real but the ratings are synthetic. Some explanation on this would be helpful. Overall, I think that a slightly worse result (in terms of imperfect recovery), but allowing a richer model (e.g., more groups/clusters, not necessarily perfect alignment between the graph and the matrix) would be a welcome addition to the paper.

Correctness: Claims and theorems seem correct to me.

Clarity: Paper is quite well written.

Relation to Prior Work: Yes

Reproducibility: Yes

Additional Feedback: Do you have any experiments by running the algorithms discussed to real ratings data? (Having hypothetical ratings but real graph seems like a validation of the graph clustering subroutines rather than ratings..)


Review 2

Summary and Contributions: This paper proposes a matrix completion algorithm for binary matrix with graph side information. While this problem has been studied before, the paper assumes there is group structure inside the cluster. The paper gives the sample complexity for such kind of problem, depending on the graph structure and the rate of observation. The paper also proposes an algorithm for solving such kind of algorithm, with a theoretical guarantee. Empirical results on synthetic and semi-synthetic data are used.

Strengths: The paper proposes a new structure of graph side information. Rigorous theoretical bound are provided for using such new structure of graph side information. They also propose an algorithm which are shown to be both theoretically and empirically tractable.

Weaknesses: The new structure needs a motivation. The problem of binary matrix completion with graph side information is already been studied. The novelty of the current paper is that in addition to the classical clustering structure, it assumes one-layer of clustering: cluster within cluster. How can such a new structure agree with the real application? I see from the experimental part no experiments on real data is given. Does this mean this problem is theoretically interesting but has few practical meanings? It is not clear why using such a performance measurement in the paper. In Eq.1, \psi is used to estimate the matrix. So, the worse-case means there is no ground truth for the target matrix? Or there is no constraint on the ground truth for the target matrix M? Such a part needs more explanation to make the metric reasonable. More discussions are needed on how the current theoretical guarantee on sample complexity compared to previous ones. Otherwise, we can not sure the new structure information in the graph do help completion. -------------------------------------------------- Not all questions have been successfully addressed (such as experimental on real data and a motivation for the problem regarding "binary" matrix completion). But I still think this paper can be accepted. I will keep my score.

Correctness: Correct

Clarity: It is generally clear. Some more explanations are need. See "Weaknessses"

Relation to Prior Work: More discussions are need. See "Weaknessses"

Reproducibility: Yes

Additional Feedback:


Review 3

Summary and Contributions: The authors consider a matrix completion problem with graph side information, where the graph forms a hierarchy. They consider the stochastic block model for analysis. Under a specific observation model and structure on the graph, the authors characterize the recovery probability of the underlying low rank matrix. They also present an algorithm that is guaranteed to obtain the right solutions under the same assumptions. The authors experimentally verify that the proposed method achieves superior results compared to a few baselines.

Strengths: Well written paper, and the theoretical results are well explained with intuition. Both an information theoretic bound as well as an algorithm that achieves the correct answer when the bounds are satisfied are presented. I'm not completely aware of some of the related works, so it's hard to tell how truly novel the theoretical results are. The experimental results are encouraging, but i'd prefer some runtime plots too.

Weaknesses: The problem setting seems contrived. Why 2 groups within each cluster specifically? can this be generalized? the polynomial time dependence in (n) for the method is potentially prohibitive. Are there ways this can be addressed? Algorithmic runtime is not discussed. Especially, in comparison to other methods that use graph structured information and are highly scalable, and in light of the poly(n) complexity, this is something that should be addressed.

Correctness: Yes

Clarity: Yes

Relation to Prior Work: Couple of references can be added in the context of using graph side information: "Collaborative filtering with graph information: Consistency and scalable methods." Advances in neural information processing systems. 2015. "Kernelized probabilistic matrix factorization: Exploiting graphs and side information." Proceedings of the 2012 SIAM international Conference on Data mining. Society for Industrial and Applied Mathematics, 2012.

Reproducibility: Yes

Additional Feedback: Line 64: not sure i follow the distinction from prior work: if clustering is a consequence of matrix completion, does it not mean matrix completion is a necessary condition? how is that more relaxed? Line 90: the problem setting seems contrived. Why 2 basis vectors specifically? can this be generalized? What happens instead, if we have 'k' bases and hence (k+1) groups? what if there are more clusters? eqn 1: considering the observations can be noisy, are you considering exact matrix recovery in the limit? if so, it will be good to call out. remark 1: This is interesting for another reason. The sample complexity for matrix recovery is O(m logm). So it seems that when the graph can be perfectly clustered, there is no advantage in using the graph? isn't this counterintuitive? I'd assume at the very least, you can solve 'K' separate matrix completion problems when the graph can be separated into K clusters. line 220: why not map +1 to +1 and 0 to -1 ? Also, why not just assume you get to see Z instead of Y in the problem formulation? Line 302: can you provide details about the subgraph? what was m, n? how many samples?


Review 4

Summary and Contributions: This paper considers a binary matrix completion problem with side information and proposes a hierarchical clustering method with refinements to achieve efficient and accurate matrix completion. The information-theoretic limit on the number of observed matrix entries is provided. Empirical analyses on synthetic and semi-real datasets showed that the proposed method can outperform several baseline methods.

Strengths: The theoretical contribution of this paper is to provide the optimal sample complexity bound for exact recovery of the binary matrix completion problem. The proposed method is not limited to recommendation problem but is also applicable to clustering problems with side information.

Weaknesses: The problem setting of this paper seems to be impractical. The authors assumed that the users are divided into two clusters and each cluster only contains three groups. This may not be a very practical setting for real recommender systems. The experiments are conducted on synthetic datasets. Although some experiments are conducted on the Poliblog Dataset, the ratings are synthetic. It will be more interesting to see how the performance of the proposed method differ from state-of-the-art recommendation algorithms on real recommendation tasks. ------------------------- I have read the authors' rebuttal and will keep my original scores for this paper.

Correctness: The theoretical and empirical results seem correct to me.

Clarity: The paper is well written.

Relation to Prior Work: Differences from prior works are clearly discussed in this paper.

Reproducibility: Yes

Additional Feedback:

[Author Response · NeurIPS 2020]

We would like to express our sincere gratitude to the reviewers for providing their valuable feedback. We are able to
collectively address only major comments below, but we will thoroughly implement all the comments in a revision.
**[R1,R2,R3,R4]-1** (*Generalization to C clusters and G groups*): In fact, we could derive the minimal sample complexity
for the generalized setting, although not included in the current draft for illustrative purpose in light of space limitation:

$$ mnp^\star = \frac{1}{(\sqrt{1-\theta}-\sqrt{\theta})^2} \max\left\{ \frac{GC}{G-R+1}m\log m, \frac{n\log n - \frac{n^2}{GC}I_g}{\delta_g}, \frac{n\log n - \frac{n^2}{GC}I_{c1} - \frac{(G-1)n^2}{GC}I_{c2}}{\delta_c} \right\}, \tag{1} $$

where the set of $G$ rating vectors in each cluster are spanned by any subset of $R$ vectors in the same set. Note that for
$(C,G,R)=(2,3,2)$, the bound in (1) reduces to the result of Thm. 1. This generalization will be added to the revision.
**[R1,R2,R3,R4]-2** (*Experiments are conducted on real graphs yet on synthetic ratings*): We used this real-synthetic
mixed dataset only for the purpose of corroborating our theory at least under real-graph settings, as in [39, 41]. However,
it can also be evaluated on purely real data settings (as suggested by R4) by slightly modifying some components in our
4-phase algorithm. For instance, we could actually make a slight change intended for a more realistic setting in which
ratings are real and noise is Gaussian (see **[R1]** below for details), and found this modified algorithm working well for
the realistic setting. We will clarify this point together with further experiments on purely real datasets in a revision.
**[R1,R2,R3]** (*Improvement over [39, 40] offered by exploiting the hierarchical graph structure; Part 1*): Remark 1
focuses on the perfect clustering/grouping regime in which user affiliations are successfully revealed. Even in this
regime, we still need to estimate four rating vectors $(v_1^A, v_2^A, v_1^B, v_2^B)$. Hence, as R3 assumed, the problem boils
down to four separate subproblems *if the hierarchical structure is ignored*. Notice that under *random sampling* of our
assumption, the recovery of each vector of length $m$ requires $m\log m$ observations due to the coupon-collecting effect,
yielding to $4m\log m$ samples. This can readily be obtained by [39, 40] which do not exploit the *hierarchical structure*.
One key observation here is that some measurements associated with $(v_3^A, v_3^B)$ are completely ignored although they
can serve to decode $(v_1^A, v_2^A, v_1^B, v_2^B)$. For example, one can decode $(v_1^A, v_2^A, v_3^A)$ *only with any two* of the three vectors
due to the linear dependency $v_3^A = v_1^A \oplus v_2^A$, which forms the basis of the *hierarchical structure*. This is exactly
what our information-theoretic results and a corresponding efficient algorithm exploit. We found this exploitation is
translated to $\frac{4}{3}$ improvement, thus yielding $3m\log m$ sample complexity. We will provide this discussion in a revision.
**[R1,R2]** (*Improvement over [39, 40] offered by exploiting the hierarchical graph structure; Part 2*): Remark 3 focuses
on the limited-clustering regime in which the hierarchical graph information is scarce. The corresponding optimal
sample complexity, that reads as $(n\log n - \frac{1}{6}n^2 I_{c1} - \frac{1}{3}n^2 I_{c2})/\delta_c$, cannot be retrieved from [39,40] since hierarchical
graph structure is not exploited in these works. We will provide further details in a revision.
**[R2,R4]** (*Motivation of hierarchical clustering in recommender systems*): In real-world recommender systems, both
item preferences and user preferences are shown to exhibit hierarchical structures [1]. For instance, users within the same
cluster can be further divided into sub-clusters (groups) with similar ratings. We will mention this in a revision.
**[R1]** (*Intuition behind the XOR dependency among rating vectors*): As you may imagine, we adopt this simplified finite-
field model only for the purpose of making an initial step towards a more generalized and realistic model. Fortunately,
characterizing the optimal sample complexity under the simple model could also shed insights into developing a
*universal and model-free* algorithm that is pertinent to any problem setting as long as some slight modification is
made. In order to demonstrate this, we now considered a practical scenario in which ratings are real (for which linear
dependency between rating vectors is well-accepted) and observation noise is Gaussian. In this setting, the *detection*
problem (under the current model) will be replaced by an *estimation* problem. So we update Alg. 1 to incorporate an
MLE of the rating vectors; and modify the local refinement criterion on Line 8 in Alg. 2 to find the group that minimizes
some properly-defined distance metric between the observed and estimated ratings. We also conducted an experiment
(similar to Fig. 2-d) on semi-real data under real field and Gaussian noise, and we found our algorithm still achieves
superior performance over the state of the arts. We will include all of these in a revision.
**[R2]** (*Re. the worst-case error probability*): $\mathcal{M}^{(\delta)}$ is the set of ground-truth matrices $M$ subject to $\delta := \{\delta_g, \delta_c\}$. Hence,
there exists indeed ground-truth for the target matrix. Since the error probability may vary depending on different
choices of $M$ (some matrices may be harder to estimate), we employ a conventional minimax approach wherein the
goal is to minimize the maximum error probability. For clarification, we will elaborate on this in a revision.
**[R3]-1** (*Poly-time complexity of our algorithm and its runtime*): In fact, we demonstrated in supplementary (Sec. 4.3,
Page 19) that our algorithm runs faster over state-of-the-arts, with few exceptions. We will move this important part
to the main body in a revision. Please note that the complexity bottleneck is in Phase 1 (exact clustering), as it relies
upon [54], exhibiting poly($n$) runtime. Recently, we have improved our algorithm so as to work optimally even under
almost exact (i.e. weak) clustering, yielding $O(|E|\log n)$ runtime [76]. In return, we modified Phase 4 so that the local
iterative refinement is applied on cluster affiliation, as well as group affiliation and rating vectors. Hence, the improved
overall runtime now reads $O((|\Omega| + |E|)\log n)$. The details of the improved algorithm will be provided in a revision.
**[R3]-2** (*Missing references*): Thanks for pointing out the two related papers. We will cite them in a revision.
**[R3]-3** (*Mapping between $Y$ and $Z$*): The adopted mapping (0 to +1, +1 to -1, and * to 0) is one standard way. One can
also use another mapping, as you suggested. Yes, we can start with $Z$ instead of $Y$. We will do so in a revision.

## Footnotes

[1]S. Wang, J. Tang, Y. Wang, and H. Liu, "Exploring implicit hierarchical structures for recommender systems," IJCAI, 2015.


[Meta-Review · NeurIPS 2020]

The reviewers were split on this paper. On the positive side the authors have given an algorithm that takes advantage of side information in a provable way that can lead to perfect recovery with a novel algorithm. On the negative side their were concerns about the synthetic nature of experiments as well as brittleness of theoretical set-up. Overall, I suggest that this paper should be accepted for its provable advance in its utilization of side-information for matrix recovery.